# An evolutionarily conserved metabolite inhibits biofilm formation in *Escherichia coli* K-12

Jingzhe Guo [1], Wilhelmina T Van De Ven[1], Aleksandra Skirycz[2,3,4], Venkatesh P. Thirumalaikumar[2,7], Liping Zeng[1], Quanqing Zhang [5], Gerd Ulrich Balcke[6], Alain Tissier [6] & Katayoon Dehesh [1] ✉

Methylerythritol cyclodiphosphate (MEcPP) is an intermediate in the biosynthesis of isoprenoids in plant plastids and in bacteria, and acts as a stress signal in plants. Here, we show that MEcPP regulates biofilm formation in *Escherichia coli* K-12 MG1655. Increased MEcPP levels, triggered by genetic manipulation or oxidative stress, inhibit biofilm development and production of fimbriae. Deletion of *fimE*, encoding a protein known to downregulate production of adhesive fimbriae, restores biofilm formation in cells with elevated MEcPP levels. Limited proteolysis-coupled mass spectrometry (LiP-MS) reveals that MEcPP interacts with the global regulatory protein H-NS, which is known to repress transcription of *fimE*. MEcPP prevents the binding of H-NS to the *fimE* promoter. Therefore, our results indicate that MEcPP can regulate biofilm formation by modulating H-NS activity and thus reducing fimbriae production. Further research is needed to test whether MEcPP plays similar regulatory roles in other bacteria.

As single-cell organisms, bacteria commonly utilize biofilm formation as a survival mechanism against environmental stressors. Biofilms, complex communities of cells surrounded by extracellular polymeric substances, are intricately regulated by a combination of environmental cues and cellular signaling molecules[1]. While considerable attention has been directed towards specialized signaling molecules such as cyclic diguanylate monophosphate (c-di-GMP)[2–5] and quorum sensing molecules[4,6,7], our study unveils an unforeseen regulatory role for the metabolic intermediate methylerythritol cyclodiphosphate (MEcPP) in disrupting biofilm formation through reducing fimbriae production by releasing repression on *fimE* transcription.

The *fim* locus in *Escherichia coli* includes several genes involved in the production of type 1 fimbriae, which are surface structures critical for adhesion and biofilm formation. FimE and FimB are site-specific recombinases that mediate the inversion of the *fimS* element, thereby controlling the phase variation of the *fimA* promoter. FimE primarily promotes the transition from the ON to OFF phase, reducing fimbriae production, while FimB can switch *fimA* in both directions, maintaining a dynamic balance in fimbrial expression[8,9].

The complex network of metabolic pathways encompasses roles beyond basic biochemical reactions. Intermediates and end products can serve as pivotal signaling molecules, communicating crucial information, modulating transcriptional landscapes, and ultimately coordinating metabolic and physiological processes essential for organismal survival in response to environmental stimuli. Among these, MEcPP stands out as a prominent metabolite[10,11], serving as an intermediate within the methylerythritol phosphate (MEP) pathway (Fig. 1a). This pathway is indispensable for the biosynthesis of isoprenoids (IPPs) and is utilized across a spectrum of life forms,

[1]Institute for Integrative Genome Biology and Department of Botany and Plant Sciences, University of California, Riverside, Riverside, CA, USA. [2]Boyce Thompson Institute, Ithaca, NY, USA. [3]Cornell University, Ithaca, NY, USA. [4]Michigan State University, East Lansing, MI, USA. [5]Institute for Integrative Genome Biology, Proteomics Core, University of California, Riverside, Riverside, CA, USA. [6]Leibniz Institute of Plant Biochemistry, Department of Cell and Metabolic Biology; Weinberg 3, Halle (Saale), Germany. [7]Present address: Bindley Bioscience Center, Purdue University; West Lafayette, Indiana, USA. ✉e-mail: kdehesh@ucr.edu

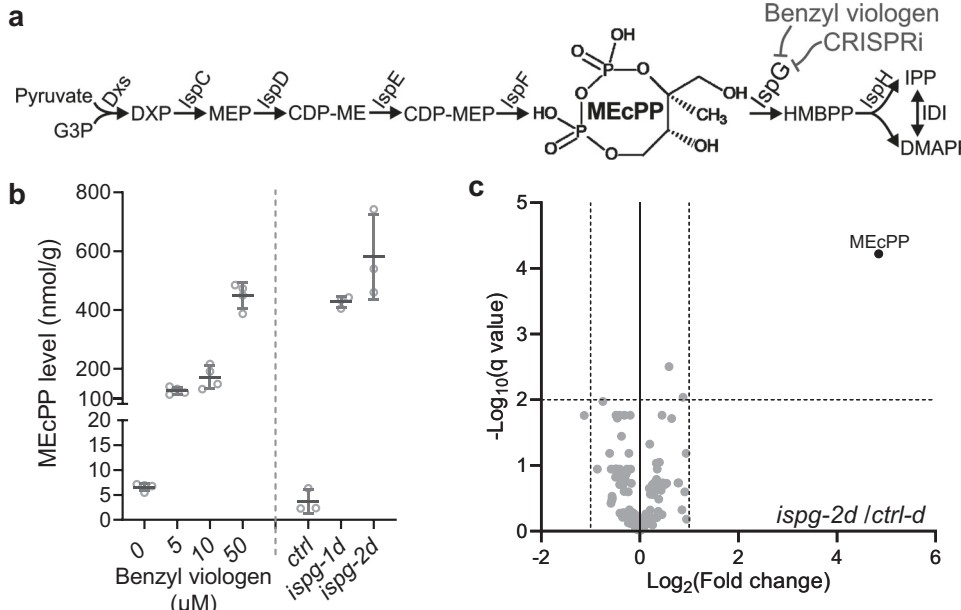

**Fig. 1 | Enhanced MEcPP levels in *E. coli* via genetic manipulation and oxidative stress. a** Simplified schematics of the MEP pathway in *E. coli*, demonstrating the accumulation of MEcPP, with chemical structure superimposed, using two methods: benzyl viologen application to inhibit IspG activity or CRISPRi-mediated reduction of *ispG* expression. **b** Measurement of MEcPP concentrations in *E. coli* exposed to varying concentrations of benzyl viologen or CRISPRi constructs targeting *ispG* (*ispg-1d*, *ispg-2d*), compared to a control (*ctrl-d*). Scatter dots depict raw data points for 3 to 4 independent biological replicates per sample group. The overlaid horizontal lines indicate the mean value and the error bars show the standard deviation. **c** Volcano plot analysis displaying fold changes in metabolite abundance within the central carbon and energy metabolism of the MEcPP-enriched strain (*ispg-2d*) versus the control (*ctrl-d*). MEcPP (depicted as a black dot) is identified as the sole significantly altered metabolite through unpaired *t* tests (two-tailed). Source data are provided in a Source Data file.

spanning from Gram-negative bacteria to plastid-containing organisms like plants and apicomplexans, which acquired it through endosymbiosis[12–14].

Previous discoveries highlighting MEcPP's dual function as both an IPP precursor and a stress-responsive signaling metabolite in plants prompted us to investigate its potential signaling roles in bacteria under stress conditions. Our research has uncovered how MEcPP interacts with the histone-like nucleoid structuring (H-NS) protein, a major component of the folded chromosome in *E. coli* and related bacteria. H-NS plays a crucial role in genome evolution, DNA condensation, and transcription regulation[15]. We demonstrate that MEcPP disrupts the association between H-NS and the *fimE* promoter, thereby releasing repression on *fimE* transcription. This cascade results in decreased fimbriae production, ultimately leading to the disruption of biofilm formation.

## Results

### Deciphering MEcPP signaling dynamics in *E. coli*

Given the conserved nature of the MEP-pathway across bacteria and plastid-bearing organisms[12–14] we hypothesized that MEcPP might function as a signaling metabolite in bacteria under oxidative stress, similar to its role in plants, because the enzyme hydroxy-2-methyl-2-butenyl 4-diphosphate synthase (HDS, or GcpE/IspG), which catalyzes the conversion of MEcPP to HMBPP, has its activity compromised by the oxidation of its iron-sulfur cluster[16,17]. To investigate this hypothesis, we exposed the model organism *E. coli* to benzyl viologen (BV), a pharmacological agent known to induce oxidative stress, with the aim of enhancing MEcPP levels. Thus, we cultured *E. coli* MG1655 in an LB medium containing varying concentrations of BV, revealing a clear concordance between MEcPP levels and BV concentration. However, this treatment also reduced *E. coli* growth at 37 °C, particularly evident with 50 μM BV, highlighting the anticipated multifaceted impact of BV on other oxidation-sensitive enzyme activities (Fig. 1b and Supplementary Fig. 1).

To mitigate the broad effects of BV, we employed a genetic strategy utilizing CRISPR interference (CRISPRi) to downregulate *ispG* expression. Specifically, we utilized two independent single guide RNAs (sgRNAs) targeting the non-template strand of the *ispG* coding region in the CRISPRi system. This led to effective suppression of *ispG* expression in both genetically altered strains (*ispg-1d* and *ispg-2d*), exhibiting reduced expression levels to approximately 19.4% and 4.3%, respectively, compared to the control strain (*ctrl-d*) (Supplementary Fig. 2).

As anticipated, both *ispg-1d* and *ispg-2d* strains accumulated significant levels of MEcPP, with *ispg-2d* exhibiting a more pronounced knockdown effect on *ispG*, accompanied by slightly higher MEcPP levels compared to *ispg-1d*. This mirrors observations similar to those in the MG1655 strain treated with 50 μM BV, albeit without growth impairment (Fig. 1b and Supplementary Fig. 1). This suggests that while IspG is essential in bacteria, the altered flux through the MEP pathway does not hinder bacterial growth under standard conditions at 37 °C.

To assess whether the genetic modifications induced notable metabolic changes, we conducted targeted metabolomic analyses focusing on central carbon and energy metabolism (CCEM) in the CRISPRi strain *ispg-2d* and the control (*ctrl-d*). Remarkably, MEcPP was the only metabolite significantly accumulated among the 186 CCEM metabolites in *ispg-2d* compared to *ctrl-d* (Fig. 1c and Supplementary Data 1). Thus, CRISPRi-mediated *ispG* knockdown in *E. coli* offers a robust method for elevating MEcPP levels without detectable changes in growth or the levels of other metabolites, facilitating investigations into MEcPP-specific alterations in the transcriptomic landscape and their associated physiological consequences.

### MEcPP levels regulate biofilm dynamics in *E. coli*

The activation of general stress response genes represents one of the earliest detectable events in plants undergoing MEcPP accumulation[18]. This prompted us to examine the transcriptional landscape following MEcPP accumulation in *E. coli* cells using RNA-seq analysis conducted

on the *ispg-2d* and *ctrl-d* strains to identify differentially expressed genes in response to elevated MEcPP levels. Gene Ontology enrichment analysis of genes with altered expression (≥ 2-fold change) revealed significant enrichment in categories related to biofilm formation, underscoring MEcPP's regulatory role (Supplementary Fig. 3). Intrigued by this revelation, we examined the biofilm production of CRISPRi strains cultured at room temperature and uncovered a significant decrease in biofilm formation in both *ispg-1d/−2d* strains compared to the *ctrl-d*. Surprisingly, this was accompanied by a modest reduction in growth in high MEcPP strains, in contrast to the growth data obtained from cells grown at 37 °C (Fig. 2a, c, and d). Nevertheless, despite a slight decline in cell growth compared to the control, the normalized growth versus biofilm formation still indicated reduced biofilm production in cells with heightened MEcPP levels. In addition, despite observing similar reductions in growth in both *ispg-1d/−2d* strains, the *ispg-2d* strain, exhibiting slightly higher MEcPP levels, demonstrated a more pronounced suppression of biofilm production compared to the *ispg-1d* strain. These data suggest MEcPP-dependent regulation of biofilm formation. This led us to explore whether BV-induced MEcPP accumulation could similarly affect biofilm formation. Examination of biofilm production in *E. coli* MG1655 cells treated with 5 μM and 10 μM BV revealed a reduction in biofilm formation, positively correlated with the level of accumulated MEcPP (Figs. 1b, 2b, c, and d). We refrained from using 50 μM BV treatment due to its substantial growth suppression, despite resulting in a similar level of MEcPP accumulation compared to the CRISPRi strains.

These findings underscore an inverse correlation between MEcPP levels and biofilm formation, revealing the intricate interplay between MEcPP accumulation and bacterial biofilm dynamics.

## Transposon screening uncovers MEcPP's biofilm regulatory role via FimE

To uncover the molecular mechanisms behind MEcPP-mediated biofilm suppression, we conducted a revertant screening to identify strains capable of restoring biofilm production despite elevated MEcPP levels. Toward this, we employed an IPTG-controlled conditional suicide plasmid-mediated transposon-insertion mutagenesis system[19], initially consolidating the two plasmids from the *ispg-2d* strain of the CRISPRi system (containing dCAS9 and the sgRNA expression cassette individually) into a single plasmid. This amalgamation gave rise to the 2-in-1 CRISPRi strain, *ispg-2*, which exhibited the high MEcPP accumulation phenotype and biofilm suppression akin to the *ispg-2d* strain (Supplementary Fig. 4). Utilizing this modified strain, we generated a transposon-insertion mutant library and screened for revertants capable of reinstating biofilm production.

Among the identified revertants, we pinpointed one harboring a transposon insertion in the coding region of *fimE* (Fig. 3a). FimE, one of the two DNA recombinases, regulates the phase transition of the *fimA* promoter, controlling the production of type 1 fimbriae appendices on the outer membrane of *E. coli* cells[20–28]. FimE plays a pivotal role in mediating the ON-to-OFF transition of the *fimA* promoter[20,25,26,28] and is recognized as a key negative regulator of biofilm formation (Supplementary Fig. 5)[29,30]. Notably, independent knockout of *fimE* via homologous recombination in the *ispg-2* strain effectively restored biofilm production, while overexpression of *fimE* in the revertant background completely abolished biofilm (Fig. 3a and Supplementary Fig. 6). These findings strongly implicate FimE as a critical molecular component in MEcPP-mediated biofilm suppression.

To validate the involvement of FimE, and type 1 fimbriae production in *E. coli* cells with elevated MEcPP levels, we directly visualized fimbriae using atomic force microscopy (AFM) in various strains. Specifically, our observation revealed that the *ispg-2* strain, characterized by elevated MEcPP levels, exhibited a significantly lower percentage of fimbriae-producing cells compared to the control strain,

whereas the percentage in the revertant strain exhibited substantial recovery (Fig. 3b, d). Similarly, the percentage of fimbriae-producing cells decreased with increasing concentrations of BV treatment (Fig. 3c, e). These results strongly suggest that MEcPP-mediated suppression of fimbriae production through the regulation of *fimE* contributed to the observed reduction in biofilm formation.

## MEcPP interacts with H-NS, a transcriptional repressor of *fimE*

To gain insight into MEcPP's role in regulating type 1 fimbriae production, we investigated potential metabolite-protein interactions using limited proteolysis-coupled mass spectrometry (LiP-MS). Specifically, we compared the *ispg-2d* strain, characterized by elevated MEcPP levels, with the *ctrl-d* strain displaying normal MEcPP levels (Fig. 4a).

The reliability of the LiP-MS assay was validated through the detection of peptides located within the recognized MEcPP-binding site of the IspG enzyme, responsible for catalyzing the conversion of MEcPP to HMBPP. Furthermore, among the array of potential MEcPP-binding candidates, our investigation identified H-NS, a well-established transcriptional repressor of *fimE* (Fig. 4a). Specifically, we established that MEcPP binds at the dimerization site of H-NS, which is essential for its gene-silencing activity[31]. The recurrent detection of conserved peptide characteristics from H-NS across numerous distinct LiP-MS assays, notably those originating from the analysis of wild-type total protein treated in vitro with 200 μM MEcPP or MEP (control), significantly bolsters the credibility of our findings. (Fig. 4b). To verify the authenticity and specificity of the binding between MEcPP and H-NS, we conducted a protein thermal shift assay. Our results showed that while MEcPP consistently altered the melting temperature of H-NS, its analogs HMBPP (Fig. 4c) and MEP (Supplementary Fig. 7) did not elicit a similar effect. Furthermore, MEcPP and HMBPP, two known IspG binding metabolites[32,33], effectively changed the melting temperature of IspG, but not that of the negative control IspC enzyme (Fig. 4c).

Moreover, we quantified the binding affinity between MEcPP and H-NS as 6.46 ± 2.17 μM using microscale thermophoresis (Fig. 4d). As a control, we determined the binding affinity between our purified recombinant IspG protein and its substrate MEcPP, yielding a measurement of 29.42 ± 10.00 μM (Fig. 4e).

In essence, our results provide compelling evidence of the interaction between MEcPP and H-NS, the transcriptional repressor of *fimE*, thereby elucidating MEcPP's role in regulating type 1 fimbriae production.

## MEcPP prevents H-NS from binding to the *fimE* promoter

Given MEcPP's interaction with H-NS, we investigated its impact on the transcriptional regulation of *fimE*. H-NS functions as a transcriptional repressor of *fimE* by binding to its promoter. Therefore, we initially employed the electrophoretic mobility shift assay (EMSA) and confirmed that pre-incubation of MEcPP with H-NS is necessary to inhibit H-NS binding to *fimE* promoter. However, this inhibition proved ineffective when H-NS was already bound to the promoter DNA (Supplementary Fig. 8). Utilizing this in vitro binding criterion, we subsequently demonstrated a concentration-dependent decrease in the fraction of the *fimE* promoter bound by H-NS upon MEcPP addition, whereas MEP failed to induce similar alterations in the mobility behavior of the H-NS-bound *fimE* promoter (Supplementary Fig. 9). The data suggests that in vivo, accumulated MEcPP under oxidative stress is capable of binding to H-NS and impeding its binding to the *fimE* promoter.

The reduction in H-NS association with the *fimE* promoter suggests a derepression of *fimE* transcription. Indeed, we observed significantly increased *fimE* expression in both the high-MEcPP accumulating *ispg-2* strain and BV-treated *E. coli* cells using quantitative PCR (qPCR) (Fig. 5a, b). The elevated *fimE* expression likely led to

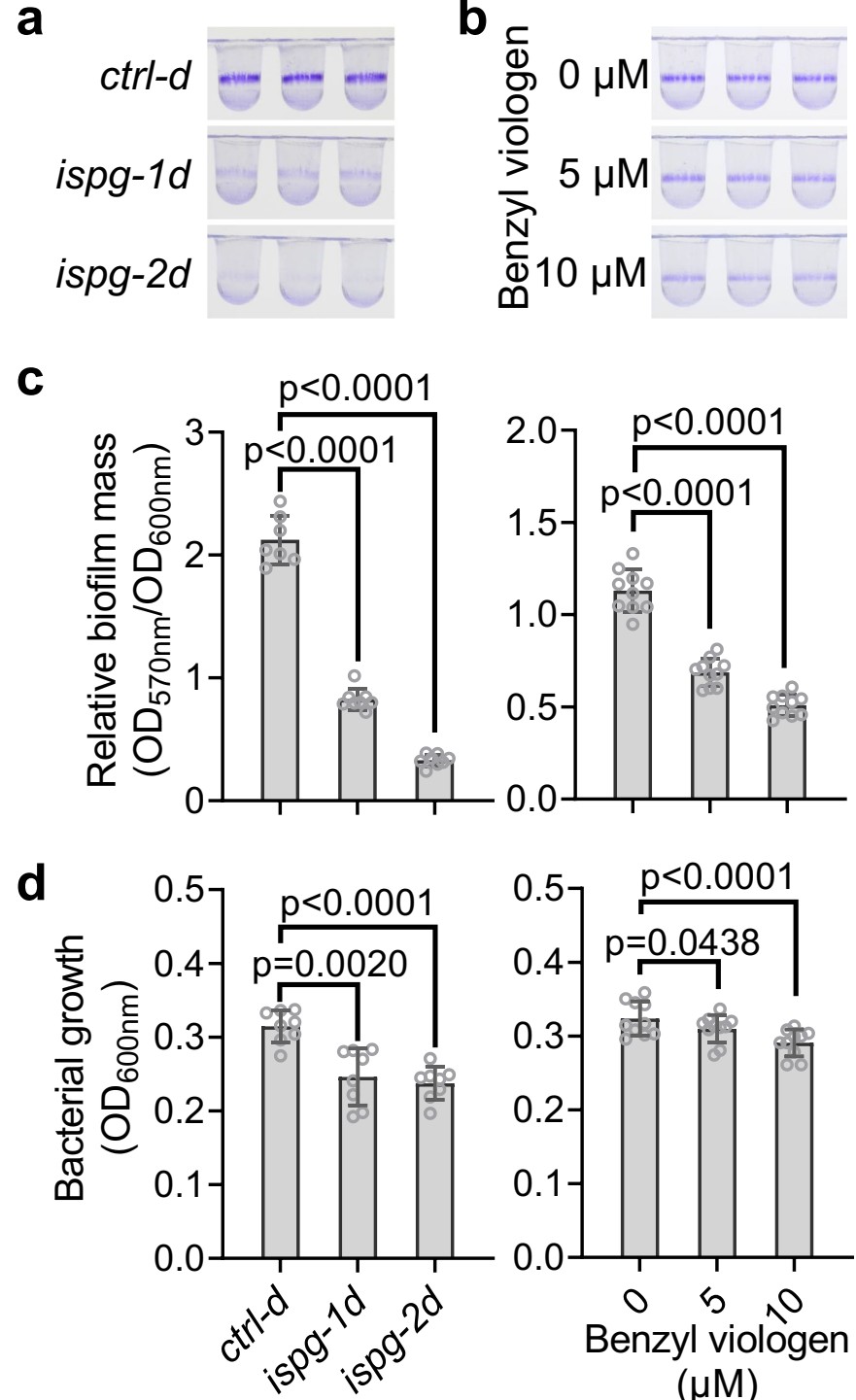

**Fig. 2 | Inverse relationship between MEcPP levels and biofilm formation.**
**a**, **b** Visualization of biofilms produced by CRISPRi strains (**a**), and MG1655 cells statically grown at room temperature in media with increasing concentrations of benzyl viologen (**b**), using crystal violet staining. **c**, **d** Analysis of relative biofilm mass calculated by the ratio between the $OD_{570nm}$ of dissolved crystal violet-stained biofilm and the $OD_{600nm}$ of planktonic cells at the time of staining (**c**) and bacterial growth measured at $OD_{600nm}$ (**d**). The results are derived from three biological replicates per genotype or treatment, and each biological replicate includes at least two technical replicates. Bars represent the mean value, and error bars show the standard deviations. Statistical tests used include Brown-Forsythe and Welch ANOVA and two-sided Dunnett's multiple comparisons tests (CRISPRi strains) and RM one-way ANOVA tests with the Geisser-Greenhouse correction and two-sided Dunnett's multiple comparisons tests (BV treatment). Source data are provided in a Source Data file.

the conversion of the *fimA* promoter orientation from ON to OFF phase, as FimE predominantly facilitates this transition[20,24].

To assess this hypothesis, we analyzed each bacterial population to determine the proportions of cells with the *fimA* promoter element in the ON or the OFF orientation, using PCR followed by HinfI restriction enzyme digestion. Our analysis revealed that the high-MEcPP *ispg-2* strain exhibited an increased proportion of *fimA* promoters in the OFF phase compared to the *ctrl-d* strain. Conversely, the revertant strain (*ispg-2 fimE::Tn5*) displayed an increased proportion of promoters in the ON phase, while strains overexpressing *fimE*

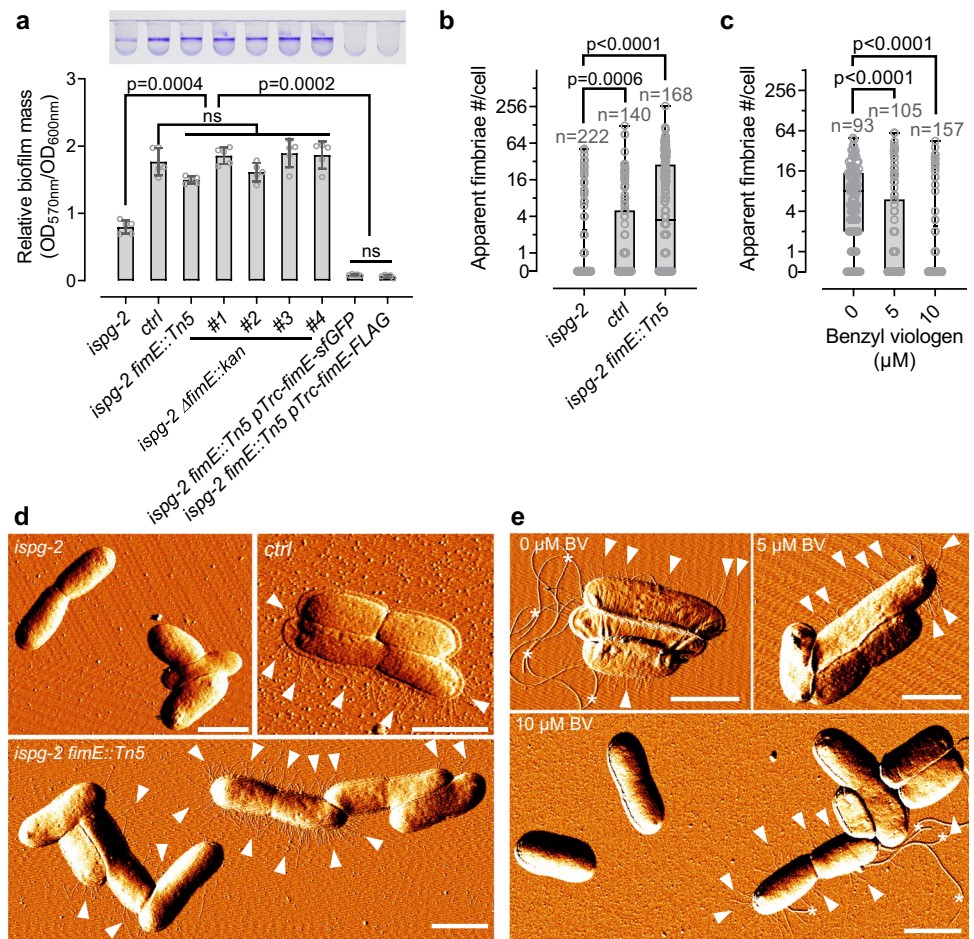

**Fig. 3 | Identification of a *fimE* mutant reverting biofilm formation despite high MEcPP levels. a** Biofilm production assessed in the high-MEcPP accumulating strain *ispg-2*, *fimE* transposon-insertion revertant *ispg-2 fimE::Tn5* and knockout mutants *ispg-2 ΔfimE::kan*, *fimE* overexpression strains *ispg-2 fimE::Tn5 pTrc-fimE-sfGFP* and *ispg-2 fimE::Tn5 pTrc-fimE-FLAG*. The upper panel displays crystal violet-visualized biofilm in PVC tubes, while the lower panel shows relative biofilm mass calculated as the ratio between the $OD_{570nm}$ of dissolved crystal violet-stained biofilm and the $OD_{600nm}$ of planktonic cells at the time of staining. The data are obtained from three biological replicates per genotype, with each biological replicate containing one or two technical replicates. Bars represent mean values, and error bars represent standard deviations. **b–e** AFM imaging and statistical analyses of fimbriae production examined in CRISPRi strains and the *ispg-2 fimE::Tn5* revertant (**d**), as well as in the MG1655 strain grown in increasing

concentrations of benzyl viologen (**e**). Panels (**b** and **c**) display box and whisker plots of the apparent number of fimbriae outside of the aforementioned *E. coli* cells. The box represents the interquartile range from the 25th to 75th percentiles, with a horizontal line inside indicating the median value; the whiskers extend to the minimum and maximum values in the data, and all individual data points are depicted as dots. The apparent number of fimbriae is counted across all cells within a minimum of six 50 × 50 μm imaging fields, using slides from two biological replicates per genotype. Statistical tests used are Brown-Forsythe and Welch ANOVA and two-sided Dunnett's multiple comparisons tests (**a**) and Kruskal-Wallis ANOVA and two-sided Dunn's multiple comparisons tests (**b**, **c**). Panels (**d** and **e**) show representative AFM images of the analyzed cells, with white arrowheads pointing at clusters of fimbriae and white asterisks denoting flagella, with scale bars indicating 2 μm. Source data are provided in a Source Data file.

(*ispg-2 fimE::Tn5 pTrc-fimE-sfGFP*) showed no promoters in the ON phase (Supplementary Fig. 6 and Fig. 5c). Similarly, MG1655 cells treated with increasing concentrations of BV exhibited a reduced fraction of *fimA* promoters in the ON phase compared to untreated cells (Fig. 5d).

In summary, our results indicate that accumulated MEcPP binds to H-NS and decreases its affinity for DNA when H-NS is not already bound to the DNA. This interaction releases H-NS from its transcriptional suppression of *fimE* expression. This upregulation of *fimE* triggers the transition of the *fimA* promoter to its OFF phase, thereby reducing fimbriae production.

## Discussion

Recent studies highlight the vital role of metabolite-protein interactions in cellular signaling under stress conditions[34,35]. MEcPP, an intermediary of the MEP pathway essential for isoprenoid synthesis, serves as a prime example of this phenomenon. Beyond its role as a

metabolic intermediate[13], MEcPP functions as a signaling molecule in plants, activating stress response genes[11,18].

Our investigation sheds light on an unforeseen aspect of MEcPP's role in regulating *E. coli* biofilm formation. Elevated MEcPP levels, whether induced by genetic manipulation or oxidative stress, disrupt biofilm formation by interacting with the dimerization site of H-NS, which is essential for its gene-silencing activity[31]. This interaction results in the derepression of *fimE* transcription, consequently reducing type 1 fimbriae production critical for biofilm formation.

MEcPP, a conserved plant metabolite, may influence a variety of H-NS-regulated pathways beyond *fimE* due to H-NS's role in regulating genes, including virulence factors. This implies that MEcPP might modulate virulence gene expression in pathogenic *E. coli* strains such as enteroaggregative *E. coli* (EAEC), uropathogenic *E. coli* (UPEC), and enterohemorrhagic *E. coli* (EHEC), potentially affecting their pathogenicity. MEcPP might also disrupt other H-NS-regulated sites within the *fim* locus, such as the *fimB* promoter and *fimS*

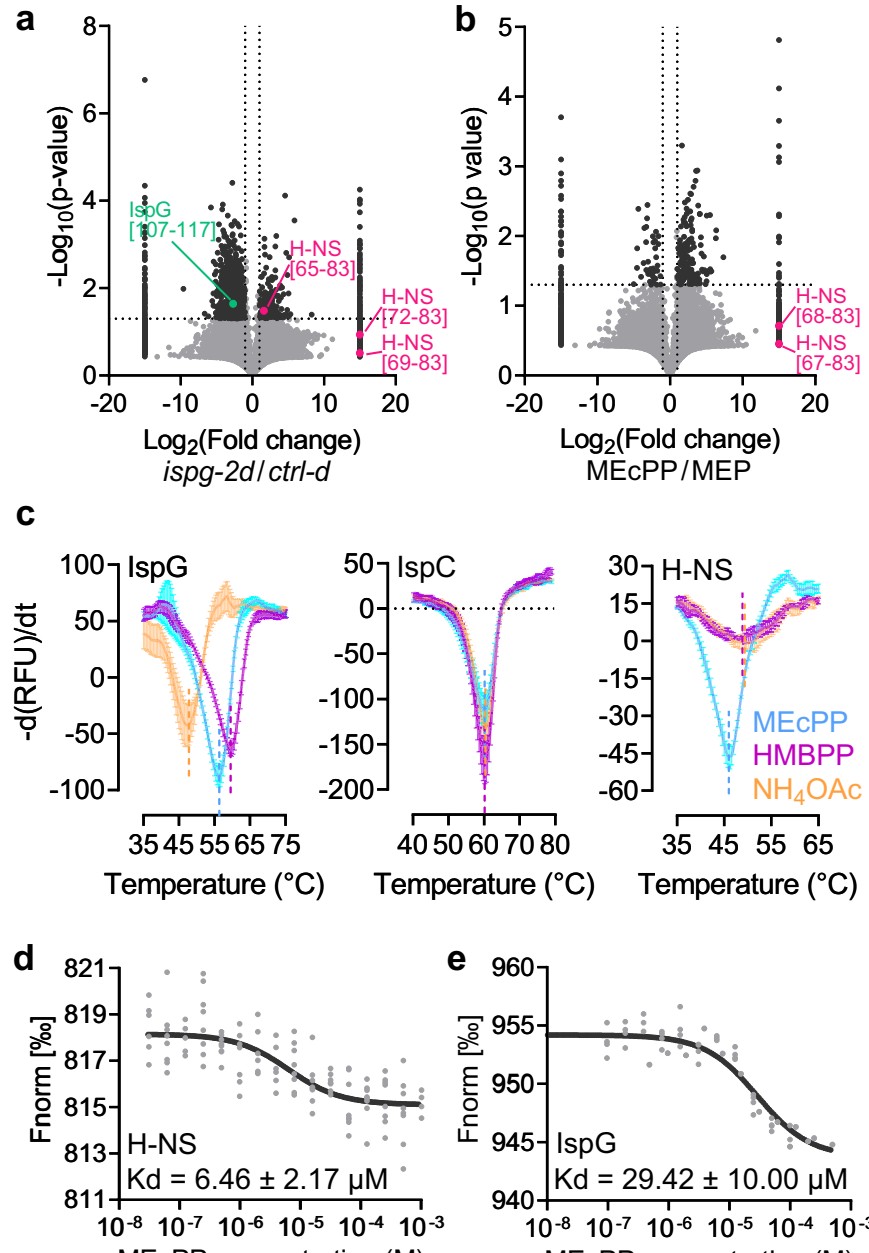

**Fig. 4 | MEcPP interacts with H-NS, a transcriptional repressor of *fimE*. a, b** Two independent LiP-MS analyses show differentially represented peptide features in the high-MEcPP *ispg-2d* and the control *ctrl-d* strains (**a**) and in wild-type *E. coli* total protein treated with MEcPP or MEP (**b**). The volcano plot displays $\log_{10}$-transformed *p*-values (two-tailed, unpaired *t* tests) against $\log_2$-transformed fold changes in normalized LiP peptide abundance between *ispg-2d* and *ctrl-d* strains (**a**), and between MEcPP- and MEP-treated total proteins (**b**). Significantly altered features (more than 2-fold change and *p* < 0.05, or nondetected in one group) are depicted in black, with purple dots highlighting overrepresented peptide features of H-NS in the high-MEcPP strain *ispg-2d* and MEcPP-treated wild-type total proteins. A green dot in (**a**) represents the underrepresented peptide feature of IspG in the *ispg-2d* strain. **c** Protein thermal shift assay demonstrates a shift in the melting temperature of H-NS in the presence of MEcPP, but not its analog HMBPP or $NH_4OAc$. IspC and IspG serve as negative and positive controls, respectively. Error bars represent the standard deviation, based on data from three technical replicates. Similar results were observed in two additional experiments. **d, e** Binding affinity determination of H-NS (**d**) and IspG (**e**) with MEcPP assessed by microscale thermophoresis. Source data are provided in a Source Data file.

element[36–39]. Its ability to prevent H-NS binding suggests a broader impact on the H-NS regulon, which includes genes involved in virulence, stress response, and metabolism. This highlights the potential of plant-derived compounds to disrupt biofilm formation and offers new therapeutic strategies against biofilm-associated infections, such as urinary tract infections. Further, multiomics and transcriptomic analyses could uncover additional pathways and genes influenced by MEcPP, providing deeper insights into its role in bacterial adaptation, survival, virulence, and resistance, particularly in H-

NS-regulated sites within the *fim* locus, such as the *fimB* promoter and the *fimS* element.

Despite appearing counterintuitive, the reduction in biofilm formation in response to MEcPP accumulation, an intermediate produced during oxidative stress, may signify a bacterial coping mechanism. In conditions of heightened oxidative stress, cells face increased susceptibility to death while also fostering the growth of biofilms[40]. However, in instances of oxidative stress where cell demise isn't immediately evident (Supplementary Fig. 1), MEcPP intervenes to

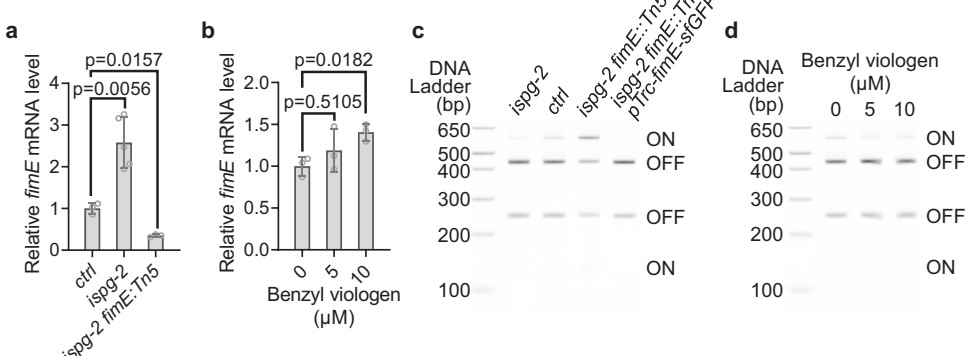

**Fig. 5 | MEcPP derepresses *fimE* transcription. a, b** Relative expression levels of *fimE* assessed in CRISPRi strains and *ispg-2 fimE::Tn5* revertant (**a**), as well as in MG1655 cells grown in increasing concentrations of benzyl viologen (**b**). Bars represent the mean value, and error bars show the standard deviations. Statistical analyses used are Brown-Forsythe and Welch ANOVA and two-sided Dunnett's multiple comparisons tests. The data are obtained from at least three biological replicates per genotype or treatment, with each biological replicate comprising two technical replicates. **c, d** Polyacrylamide gel visualization of the relative abundance of the ON and OFF phases in the *fimA* promoter in CRISPRi strains, the *ispg-2 fimE::Tn5* revertant, and the *fimE* overexpressor in the revertant (*ispg-2 fimE::Tn5 pTrc-fimE-sfGFP*) (**d**), as well as in MG1655 strains grown in increasing concentrations of benzyl viologen (**d**). HinfI restriction digestion of the amplified *fimA* promoter generates differential band patterns, visualizing the ON (520 bp + 124 bp) and OFF (405 bp and 239 bp) phases of the promoter. Consistent results in (**c** and **d**) were confirmed in 2 additional experiments. Source data are provided in a Source Data file.

impede bacterial biofilm formation. This hindrance promotes dispersion, potentially assisting in evading oxidative stress, enhancing access to antioxidants, and diminishing competition for spatial resources. As a result, individual bacteria may exhibit heightened resilience against oxidative stress.

Intriguingly, previous studies have linked MEcPP to binding with Hc1, a histone-like protein in chlamydia. This interaction caused the condensed nucleoid to revert to a relaxed conformation in *E. coli* cells overexpressing Hc1[41], suggesting a potential role for MEcPP in regulating nucleoid architecture and, by extension, adjusting DNA accessibility to the transcriptional machinery. Our study also reveals that the DNA-binding protein HU-alpha, a histone-like DNA-binding protein involved in chromatin organization[42], may be a target for MEcPP binding (Supplementary Data 4, 5 and 6). This suggests that MEcPP's influence on nucleoid architecture extends beyond its interaction with H-NS, and mirrors the role of polyamines in DNA condensation through nucleoid-associated proteins (NAPs) like HU[43]. DNA condensation via NAPs such as H-NS and HU is crucial for regulating gene expression by altering DNA topology and accessibility[44]. This regulation affects the *fimA* gene in *E. coli*, where changes in DNA supercoiling and NAP binding lead to ON-OFF switching of the *fimA* promoter, impacting type 1 fimbriae expression and bacterial adaptability[44,45]. This underscores the multifaceted role of MEcPP, showcasing its ability to modulate bacterial physiology through two distinct mechanisms: direct influence on nucleoid structure via HU-alpha interaction and regulating gene expression through interaction with H-NS.

Overall, MEcPP's engagement with histone-like proteins represents a sophisticated regulatory mechanism employed by bacteria to adapt to environmental changes and finely tune their gene expression profiles accordingly. These interactions with nucleoid-associated proteins highlight the interconnectedness of the metabolic pathways involved in maintaining nucleoid structural integrity and regulating DNA accessibility to various cellular processes.

A recent report indicated that c-di-GMP inhibits the DNA binding activity of H-NS in Salmonella[46], mirroring our discovery of increased transcripts of the c-di-GMP phosphodiesterase *pdeL* in bacteria with heightened MEcPP levels (Supplementary Data 3). This implies a potential reduction in c-di-GMP levels, a pivotal second messenger known to facilitate biofilm formation when at a high cellular level[2–5]. Both c-di-GMP and MEcPP target H-NS, raising the possibility that these molecules might act synergistically, antagonistically, or independently

in *E. coli*. Understanding these interactions could have significant implications for *E. coli* physiology, particularly in regulating biofilm dynamics and bacterial adaptability.

In conclusion, our study emphasizes the complex interplay among metabolite levels, protein interactions, and the consequences of disrupting this delicate balance on nucleoid architecture and biofilm formation. Moreover, our findings underscore the significance of the conserved MEP pathway in stress responses, particularly through the role of its intermediate, MEcPP, in both plants and bacteria. Understanding the role of MEcPP in biofilm formation opens promising avenues for developing innovative strategies to combat chronic infectious diseases and tackle biofouling challenges across various industrial settings through targeted metabolic interventions.

## Methods

### Bacteria growth conditions

*E. coli* MG1655 Z1 *malE⁻* (F⁻, lambda⁻, rph-1, lacI$^q$, P$_{N25}$tetR, SpR, malE⁻)[47] is acquired from Addgene (a gift from Keith Tyo, Addgene plasmid # 65915), herein referred to as the MG1655 strain. Unless stated otherwise, all bacteria were cultured at 37 °C in Luria Broth (Miller) (Sigma, L3522, USA) liquid medium or LB plates supplemented with 1.5% agar. The concentration of antibiotics for antibiotic-resistance selection were as follows: carbenicillin, 100 mg/L; kanamycin, 50 mg/L; gentamycin, 20 mg/L; chloramphenicol, 35 mg/L. A stock solution of 100 mM benzyl viologen (BV) was always freshly prepared and diluted into an LB medium to the desired concentration.

### CRISPR interference-mediated knock-down of *ispG* in *E. coli*

Two sgRNAs targeting the non-template strand of 5'-end *ispG* coding region are designed using the Chop-chop website[48]. A control sgRNA (*ctrl-d*) is also generated by introducing two mismatch nucleotides in the *ispg-2d* sgRNA spacer region immediately adjacent to the protospacer adjacent motif (PAM), as diminishing the binding of dCAS9 to the target DNA strand. Incorporation of the above sgRNAs into the pgRNA vector of the CRISPRi system is performed as described[49,50]. Finally, the pgRNA and pdCAS9 vectors were transformed into *E. coli* MG1655 Z1 *malE* strain by electroporation.

### Plasmid constructs and primers

Sources of used plasmids are listed in Supplementary Table 1. All plasmids generated in this study, along with the primers and assembly

methods, are listed in Supplementary Table 1. The *E. coli* strain for plasmid propagations is TOP10 (Invitrogen, ThermoFisher, USA). All primers used in the current studies are listed and annotated in Supplementary Table 2.

## LC-MS analysis of MEcPP levels

An 8 mL LB medium supplemented with appropriate antibiotics or indicated concentration of BV was inoculated with 32 μL overnight culture of CRISPRi strains or wild-type *E. coli* cells and was shaken in a 37 °C incubator shaker until $OD_{600nm}$ reached ~1.0. Bacteria pellet from 5 mL of the culture was collected by centrifuge at $2500 \times g$ for 10 min at 4 °C. The pellet was then washed with ice-cold water and pelleted into a preweighed 1.5 mL microtube for measuring the fresh weight of the collected bacteria. MEcPP was then extracted from these weighed pellets according to the published method[51]. Quantification of MEcPP levels was conducted using our established method[52] with 3–4 biological repeats for each genotype or BV treatment.

## Targeted Metabolomics of the Central Carbon and Energy Metabolism (CCEM)

Five biological repeats of *ispg-2d*, *ctrl-d* strains were cultured in LB medium supplemented with appropriate antibiotics in a 37 °C incubator shaker till late-logarithmic phase ($OD600_{nm} \approx 0.8$). Bacteria pellets were collected and washed with deionized water by centrifugation at 4 °C, followed by lyophilization. Subsequently, 30 mg of lyophilized *E. coli* was extracted in a two-phase extraction as described[53] using 900 μL of dichloromethane/ ethanol (2:1, v/v) and 150 μL of aqueous hydrochloric acid of pH 1.4. After two rounds of extraction, aqueous extracts were combined and stored at −80 °C until analysis. Separation of hydrophilic metabolites was performed by ion-pairing chromatography on a Nucleoshell RP18 column (2.1 × 150 mm, particle size 2.1 μm, Macherey & Nagel, GmbH, Düren, Germany) using a Waters ACQUITY UPLC System, equipped with an ACQUITY Binary Solvent Manager and ACQUITY Sample Manager (5 μL injection volume; Waters GmbH, Eschborn, Germany). Eluents A and B were aqueous 10 mM tributyl amine (adjusted to pH 6.2 with glacial acetic acid) and acetonitrile, respectively. Elution was performed isocratically for 2 min with 2% eluent B, from 2 to 18 min with a linear gradient up to 36% B, and from 18–21 min up to 95% B, and isocratically from 21 min to 22.5 min with 95% B, from 22.51 to 26 min again down to 2% B. The flow rate was set to 400 μL/min, and the column temperature was maintained at 40 °C.

Mass spectrometric analyses of small molecules were performed by targeted MS/MS via multiple reaction monitoring (MRM) by using a QTRAP 6500 (AB Sciex GmbH, Darmstadt, Germany) operating in negative ionization mode and controlled by Analyst 1.7.1 (AB Sciex GmbH, Darmstadt, Germany) (Supplementary Data 2). The source operation parameters were the following: ion spray voltage, −4500 V; nebulizing gas, 60 psi; source temperature, 450 °C; drying gas, 70 psi; curtain gas, 35 psi.

Peak integration was performed using the MultiQuant software version 3.0.3 (Sciex, Toronto, CA). Individual CCEM peak areas of $n = 5$ biological replicates were normalized to the total peak area for each sample and averaged for the *ispg-2d* and the *ctrl-d* group, respectively. Finally, the group average of *ispg-2d* was divided by the average *ctrl-d* control samples. All area ratio data were logarithmized to the basis of 2.

## RNA-Seq

Total RNAs from 37 °C shaking cultured *E. coli* cells of *ispg-2d* and *ctrl-d* strains (at late-logarithmic phase, $OD_{600nm} \approx 1.0$, three biological replicates for each genotype) were extracted using an Aurum™ total RNA mini kit (Bio-Rad, USA). Genomic DNAs were removed from the total RNAs using a TURBO DNA-free Kit (Thermo Fisher, USA). Subsequently, ribosomal RNAs were removed using NEBNext rRNA

depletion Kit (Bacteria) (NEB, USA), followed by RNA-Seq library preparation using NEBNext Ultra II Directional RNA Library Prep Kit for Illumina (NEB, USA). The quality of all RNA samples and prepared RNA-seq libraries were assessed using Agilent 2100 Bioanalyzer. The RNA-seq was conducted with an Illumina NovaSeq 6000 system on a S4 flow cell. HISAT2[54] was used to align sequencing reads to the *E. coli* reference genome. DESeq2[55] was used to count and normalize mapped reads. Genes with 2-fold altered expression levels and *q*-value ≤ 0.05 (unpaired *t* tests) were identified as differentially expressed genes.

## Biofilm culture and visualization

Biofilm production was quantified using crystal violet staining, and statistical significance was determined using Brown-Forsythe and Welch ANOVA and Dunnett's multiple comparisons tests (CRISPRi strains) and RM one-way ANOVA tests with the Geisser-Greenhouse correction and Dunnett's multiple comparisons tests (BV treatment), with $p < 0.05$ considered statistically significant. Specifically, overnight cultured *E. coli* cells were 1:100 diluted into fresh LB medium with appropriate antibiotics in 96-well non-treated polystyrene (PS) plates (Greiner, Germany) or polyvinyl chloride (PVC) "U" bottom plates (Corning, USA). After 48 h of static culture at room temperature (22–23 °C), $OD_{600nm}$ of the bacterial cultures were measured on a Molecular Devices SpectraMax iD5 plate reader, followed by crystal violet staining of biofilm as previously described[56]. Stained biofilm was dissolved in 30% acetic acid and quantified by measuring $OD_{570 nm}$ on the plate reader.

## AFM imaging of *E. coli* fimbriae

Overnight-grown *E. coli* cells were 1:2000 diluted into fresh LB medium with appropriate antibiotics or indicated concentrations of BV and statically cultured in a 37 °C incubator for 24 h. The culture at medium-air interface was transferred onto poly-L-lysine-coated cover glasses and incubated at room temperature for 1 h for adhesion of bacteria cells to coverglass. Subsequently, the cover glasses were washed three times by dipping into a tank of deionized water. The cover glasses were then air-dried and mounted on a JPK NanoWizard 4a atomic force microscope (Bruker, USA) for imaging with AC mode using a Bruker RFESPG-75 or μMasch HQ:NSC18/Al BS cantilevers.

## Sample preparation for LiP-MS proteome analyses

Three biological repeats of *ispg-2d*, *ctrl-d*, and MG1655 strain were cultured in LB medium supplemented with appropriate antibiotics in a 37 °C incubator shaker till late-logarithmic phase ($OD_{600nm} \approx 0.8$). Total protein extraction was prepared according to the previously described[57]. Total protein extraction from *ispg-2d* and *ctrl-d* was directly subject to LiP-MS assay[57]. For exogenous MEcPP or MEP treatment, total protein extraction from MG1655 was incubated with 200 μM MEcPP or MEP at room temperature for 30 min, then was used for LiP-MS assays[57]. An equal input of 100 μg of all extracted or MEcPP/ MEP-treated protein samples was subject to limited proteolysis (LiP) using proteinase K followed by trypsin/Lys-C digestion or directly digested by trypsin/Lys-C. The abundance of unique peptide features was quantified in the LiP samples, while protein abundance was quantified in the sample digested with only trypsin/Lys-C, which was then used for normalization of the unique peptide abundance in the LiP samples. The protein samples were processed for proteomic analysis following the established method[57]. Briefly, the samples from LiP-MS assays were dissolved in a denaturation buffer (6 M urea, 2 M thiourea dissolved in 50 mM ammonium bicarbonate) followed by reduction with 100 μM DTT for 60 min at room temperature. The alkylation step was done by 60 minutes of incubation with 300 μM iodoacetamide in the dark. Residual iodoacetamide was quenched by 100 μM DTT for 10 min at room temperature. The residual urea is diluted with at least 6 volumes of 50 mM ammonium bicarbonate. Finally, LiP-MS fractions were digested using trypsin/Lys-C mixture

(Mass Spec Grade, Promega) for 16 h according to the manufacturer's instruction. Digested peptide samples were desalted using C18 sep-pak column plates. The eluted peptides were transferred to Eppendorf low-bind tubes and then dried using speed vac. The dried peptides were resuspended using a resuspension buffer (5% acetonitrile in 0.1% formic acid). An equal amount of the peptides was injected for LC-MS/MS analysis.

## Data acquisition for LiP-MS proteome analyses
The peptide mixtures were separated using a Dionex-ultimate 3000 nano-liquid-chromatography system (nHPLC) with an acclaim pepmap C18 column. A flow rate of 300 nL/min was used for the separation of complex peptides. The solvent A/B gradient was as follows: Being isocratic at 3% B for 17 min (including the first 10 min for loading and trap column), linearly increasing to 30% B at 125 min, linearly increasing to 40% B at 145 min, keeping at 95% B from 145.1 min to 155 min, shifting back to 3% B in 0.1 min (valve were returned to load position) and holding until 170 min. The peptide samples were sprayed using a nano-bore stainless-steel emitter (Fisher Scientific). Peptides were analyzed using an Orbitrap-Exploris-480 MS™ mass spectrometer. Data was collected using a data-dependent acquisition (DDA) mode. Standard mass spectrometer parameters were kept as described[58], HeLa digests (Pierce, Thermo Scientific, USA) have been used to monitor the retention time drift and mass accuracy of the instrument.

## LiP-MS proteome data process
Raw data were analyzed using the Proteome Discoverer (PD) (version 3.0, ThermoFisher Scientific) following the manufacturer's instructions. PD search was made using an *E. coli* protein database (UP000000625, *Escherichia coli* (strain K12) (K12/MG1655/ATCC 47076)) downloaded from Uniport. The trypsin enzyme was selected and used in semi settings. Common contaminants were compiled and added to the search. SEQUEST HT was used to assign the peptides, allowing a maximum of 2 missed tryptic cleavages. In addition, a precursor mass tolerance of 10 ppm and a fragment mass tolerance of 0.02 Da, with a minimum peptide length of 6 AAs, were selected. Cysteine-carbamidomethylation and methionine-oxidation were selected in default modifications. Label-free quantification (LFQ) based on MS1 precursor ion intensity was performed in Proteome Discoverer with a minimum Quan value threshold set to 0.0001 for unique peptides. The '3 Top N' peptides were used for area calculation. Quality control plots for the LiP-MS analyses are provided in Supplementary Data 7.

## Recombinant protein expression
Cloning (pET16b-IspC), expression, and purification of 6xHis-tagged IspC protein were previously described[59]. The coding sequence of *ispG* and *hns* were cloned into the pET-28a vector for expression of 6xHis-tagged recombinant protein in *E. coli* Rosetta (DE3) cells. Recombinant protein production is induced with 0.1 mM Isopropyl β-d-1-thiogalactopyranoside (IPTG) at room temperature for 4 h and affinity purified with TALON Metal Affinity Resin (Takara Bio, USA). Eluted recombinant proteins (elution buffer: 50 mM sodium phosphate buffer pH7.4, 300 mM NaCl, 100 mM imidazole) were buffer exchanged using Amicon Ultra-15 centrifugal filters (MilliporeSigma, USA) to remove imidazole. Eventually, 6xHis-H-NS is stored in 50 mM Tris-Cl pH7.5, 200 mM NaCl, 10% (v/v) glycerol, and 6xHis-IspG in 50 mM Tris-Cl pH7.5, 50 mM NaCl, 10% (v/v) glycerol. All recombinant proteins are stored as 30−50 μL aliquots in 200 μL microtubes and kept in a −80 °C freezer after snap-freezing in liquid nitrogen.

## Protein thermal shift assay
Protein thermal shift assay was performed in a Bio-Rad CFX96 Real-Time PCR detection system using a Protein Thermal Shift Dye Kit (ThermoFisher, USA). Thermal shift buffer for each recombinant protein was as follows: IspC, 25 mM PIPES pH7.0, 500 mM NaCl; IspG, 25 mM MOPS pH7.5, 50 mM NaCl; H-NS, 25 mM PIPES pH6.0, 50 mM NaCl. Each reaction (20 μL) contains 1x thermal shift dye, 2 μM recombinant protein, and 200 μM of MEcPP, HMBPP or $NH_4OAc$.

## Microscale thermophoresis
His-tagged H-NS protein was labeled with the RED-MALEIMIDE dye according to the manufacturer's instructions (Nanotemper, Germany). Binding was performed in standard capillaries in 25 mM MES pH 6.0 and 50 mM NaCl. MST power was set to high. Data are the mean of eight independent measurements from four independent titrations. His-tagged IspG protein was labeled with the His-Tag Labeling Kit RED-tris-NTA 2nd Generation dye according to the manufacturer's instruction (Nanotemper, Germany). Binding was performed in standard capillaries in 25 mM MOPS pH 7.5 and 50 mM NaCl. MST power was set to medium. Data are from four independent titrations. Data analysis was performed using Monolith software (Nanotemper, Germany).

## Electrophoresis mobility shift assay (EMSA)
DNA probe of *fimE* promoter for the EMSA was PCR amplified using Q5 High-Fidelity DNA polymerase (NEB, USA) and purified with a HighPrep PCR kit (MagBio Genomics, USA). In a 20 μL reaction, 240 nM 6xHis-H-NS was first mixed with increasing concentrations of MEcPP or MEP in a Tris buffer (10 mM Tris-Cl pH7.5, 10 mM $MgCl_2$, 10 mM KCl, 100 mM NaCl, 0.5 mM DTT, 5% glycerol, 0.1% (w/v) BSA, and 1 mM spermidine) and kept at 25 °C for 20 min on a Bio-Rad C1000 Touch Thermal Cycler. The reaction was incubated for another 20 min at 25 °C after the addition of 30 nM *fimE* promoter DNA fragments. The reaction mix was resolved by running in a 5% polyacrylamide gel (1xTAE) at 100 V. The gel was imaged in a Bio-Rad ChemiDoc MP Imaging System after staining in GelRed (Biotium, USA) staining solution (1xGelRed in 1xTAE, 100 mM NaCl) for 30 min.

## Screening of biofilm revertant after random transposon-insertion mutagenesis
Transposon-insertion mutant library of *ispg-2* strain was generated by mutagenesis plasmid pSNC-mTn5[19]. More than 9150 individual mutant clones were cultured and tested for biofilm formation. All biofilm revertants from the first-round screening were further validated for at least 3 more times which resulted in 265 mutant clones consistently recovered biofilm formation.

## Mapping transposon insertion site
An arbitrarily primed PCR approach[60] followed by Sanger sequencing was used to acquire the flanking sequences of the Tn5 transposon insertion site in the biofilm revertants. An initial examination of 7 randomly selected revertant clones has led to the identification of an *ispg-2 fimE::Tn5* mutant.

## Homolog recombination mediated *fimE* knockout
Kanamycin resistant cassette was PCR-amplified from plasmid pDONR221 using Q5 High-Fidelity DNA polymerase (NEB, USA) with primers designed with 5' extensions homologous to the sequences flanking the coding region of *fimE*. Homolog recombination mediated *fimE* knockout in *ispg-2* strain using plasmid pSIJ8-Gent[61] was performed as described[61,62].

## Reverse transcription and quantitative PCR (RT-qPCR)
Overnight cultures of CRISPRi strains (*ispg-2* and *ctrl*), the revertant *ispg-2 fimE::Tn5*, and MG1655 strain were 1:2000 diluted into 3 mL LB liquid medium with appropriate antibiotics or indicated concentrations of BV and were statically cultured at 37 °C for 24 h. Total RNAs from the aforementioned *E. coli* cells were extracted using a Bio-Rad Aurum total RNA mini kit. Genomic DNAs were removed from the

extracted RNAs using a TURBO DNA-free Kit (Invitrogen, Thermo Scientific, USA). Complete removal of genomic DNA contamination from the treated RNA samples was verified by the absence of PCR amplification using qPCR primers targeting *era* genes in the non-RT reactions. Subsequently, the RNAs were reverse transcribed using iScript™ Reverse Transcription Supermix (Bio-Rad, USA). To find the most stable reference genes for QPCR, we performed QPCR reactions using qPCR primer pairs for 7 reference genes (*secA, hcaT, era, dnaG, cysG, gyrB,* and *idnT*) on the RT products of 6 representative samples (include various genotypes and BV-treatments). The qPCR results were subjected to geNorm analysis in qBase+ (v3.4) software[63], which recommended an optimal normalization factor to be calculated using the geometric means of two reference genes, *hcaT* and *cysG*. Finally, qPCR of *ispG, fimE, hcaT,* and *cysG* was performed on all RT samples, and the resulting Cq value was analyzed by the qBase+ software, which generated normalized RNA level of *ispG* and *fimE* in the samples. Each genotype or BV treatment has at least three biological replicates and two qPCR technical replicates.

### Analysis of *fimA* promoter inversion
Analysis of the *fimA* promoter inversion was performed as previously described[37]. The promoter of *fimA* was amplified using NEB Taq DNA polymerase and purified using a HighPrep PCR kit (MagBio Genomics, USA). Purified *fimA* promoter fragments were digested with HinfI overnight at 37 °C and then resolved by running on a 5% polyacrylamide 1xTAE gel at 100 V. The gel was stained with 1x GelRed staining solution and imaged on a Bio-Rad ChemiDoc MP Imaging System.

### Reporting summary
Further information on research design is available in the Nature Portfolio Reporting Summary linked to this article.

## Data availability
The RNA-seq data have been deposited in the NCBI Sequence Read Archive (SRA) under accession number PRJNA1150654. The raw mass spectrometry proteomics data of the LiP-MS analyses have been deposited to the MassIVE repository (https://massive.ucsd.edu) with the dataset identifier MSV000094552. The CCEM metabolomics data are available at the MetaboLights repository under the accession number MTBLS2240. Source data are provided in this paper.

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

## Acknowledgements

This work is financially supported by the National Institute of Health (R01GM107311-8) and Dr. John W. Leibacher and Mrs. Kathy Cookson endowed chair funds awarded to K.D. The authors thank members of the Dehesh laboratory for critical comments on this work, Brian Zhang and Dr. Jinjin Li for helping with preparing the merged CRISPRi constructs and the transposon-insertion mutant library, Dr. Maria Fernanda Gomez Mendez and Haiyan Ke for the LC-MS analyses of MEcPP levels. We thank the Genomics Core Facility at the Institute for Integrative Genome Biology (IIGB), UC Riverside, for performing the RNA-seq, Manhoi Hur from the Metabolomics Core Facility (IIGB, UCR) for suggestions on the LiP-MS data analyses, and Caroline F. Plecki for mass-spectrometry sample preparation. We also thank Dr. Huatao Guo from the University of Missouri for providing us with the mutagenesis plasmid pSNC-mTn5. We acknowledge the de.NBI MASH project (W-de.NBI-011) in the German Network for Bioinformatics Infrastructure (de.NBI) and ELIXIR-DE for support with data preparation and submission to MetaboLights.

## Author contributions

J.G. and K.D. conceived and designed the study. J.G. performed project supervision, experimental design, investigation, and data analyses. W.V. generated *E. coli* strains carrying CRISPRi plasmids and prepared protein samples for LiP-MS assay. A.S. and V.P.T. conducted mass spectrometry

analyses of the LiP-MS samples. A.S. performed microscale thermophoresis assays. L.Z. analyzed RNA-seq data. Q.Z. analyzed LiP-MS data. G.U.B. and A.T. performed targeted metabolomics of central carbon and energy metabolism. K.D. provided overall project supervision. J.G. and K.D. wrote the manuscript. All authors reviewed and approved the final version of the manuscript.

## Competing interests

The authors declare no competing interests.
