## [Transparent Peer Review file · Nature Communications]

An Evolutionarily Conserved Metabolite Inhibits Biofilm Formation in *Escherichia coli* K-12

Corresponding Author: Professor Katayoon Dehesh

Version 0:

Reviewer comments:

Reviewer #1

(Remarks to the Author)

This study by Guo J et al., is supported by a host of convincing experimental data. The authors have found that a plant stress signaling metabolite referred to as MEcPPP plays a role in modulating biofilm formation of a laboratory *E. coli* strain MG1655 through regulation of intracellular *fimE* levels. Efforts to explain the molecular mechanism involved in the release of *fimE* transcription by MEcPPP identified the binding of this metabolite to H-NS, thereby disrupting the association of H-NS with the *fimE* promoter. High levels of MEcPPP yielded increased expression of *fimE*, which favored the conversion of the *fimA* ON state to the *fimA* OFF state. The role of H-NS in regulating type 1 fimbriae expression is well documented. Likewise, the contribution of type 1 fimbriae in biofilm formation has been known for many years. The finding that a plant metabolite can influence type 1 fimbriae via disruption of the binding of H-NS to the *fimE* promoter, and consequently biofilm formation is a novel one. This report opens the door to study other plant metabolites that could influence biofilm formation could have important clinical implications in pharmacology drug discovery to treat or prevent human urinary tract infections. This should be introduced in the Discussion section. In general, this is a well-written paper that reports exciting findings and the conclusions drawn are sound.

Do the authors know if MEcPPP influence other H-NS regulated genes? For example, virulence genes regulated by H-NS in pathogenic *E. coli* or other pathogens? Can MEcPPP mitigate H-NS regulation of virulence genes carried on virulence plasmids, for example in enteroaggregative *E. coli* (EAEC), uropathogenic *E. coli* (UPEC) or enterohemorrhagic *E. coli* (EHEC)? The study uses a domesticated (labmesticated) *E. coli* strain MG1655 which does not carry virulence genes and its ability to cause biofilms is not as remarkable as that of EAEC or UPEC. I feel the Discussion section should at least touch on this possibility even if no data are available at hand or in the literature.

J.A. Girón

Reviewer #2

(Remarks to the Author)

Guo provide evidence that MEcPPP, a small-molecule metabolite, interacts with the H-NS nucleoid-associated protein, preventing its binding to DNA. H-NS molecules already bound to DNA are not displaced by MEcPPP. Superficially, these findings resemble the recent discovery (Reference 35) that the small signalling molecule c-di-GMP binds to H-NS, preventing the protein from binding to DNA, but c-di-GMP cannot displace H-NS that is already bound to DNA.

H-NS affects the transcription of a large regulon of genes, usually negatively. Therefore, any molecule that interferes with the binding of H-NS to DNA might be expected to exert broad influence on the transcriptome. The authors of the present study have chosen to focus on the *fim* operon, a genetic element that encodes type 1 fimbriae. These surface appendages are responsible for mannose-sensitive haemagglutination, adhesion of the bacteria to biotic and abiotic surfaces and play a foundational role in the production of biofilm. Thus, the authors connect MEcPPP to biofilm via the type 1 fimbrial operon.

The work is conducted in *Escherichia coli*, yet the name of this bacterium does not appear in the title, the abstract, or the one-sentence summary of the paper. It will be important to correct this omission for the reasons given below.

The authors have chosen to work with *E. coli* K-12 strain MG1655, the first *E. coli* strain to have its entire genome sequence determined (PMID: 9278503). They focus on the *fimE* gene at the *fim* genetic locus, a gene that is inactive in some

commonly-used *E. coli* K-12 laboratory strains due to the presence of a *fimE::IS1* insertion mutation (Reference 22). This has implications for the extrapolation of their findings to other *E. coli* (and *Shigella*) strains. Moreover, given that the finding that c-di-GMP affects H-NS binding to DNA was made in *Salmonella* (Reference 35), and that *Salmonella* regulates type 1 fimbrial gene expression via a mechanism that is unrelated to that found in *E. coli* (PMID: 17981960), it is important to inform the reader in the title that this is an *E. coli* K-12 study, and in the abstract that it was conducted in MG1655.

The *fim* locus is insufficiently described in the Introduction. There, the reader is given the impression that relief of H-NS-mediated repression of the *fimE* gene results in a reduction in type 1 fimbrial production, (lines 57-61) implying that FimE acts as a repressor. The authors do mention (lines 144-148) that FimE is one of two recombinases that regulate the 'phase transition' of the *fimA* promoter but the molecular details are missing and the reader should not have to consult six references (16, 21, 22, 24, 25 and 26) in order to piece together information that is foundational to the study of Guo et al. This point is also important because the *fimE* gene is regulated not only by the activity of its own promoter but also at the level of FimE mRNA stability by the orientation in the chromosome of *fimS*, the genetic element that carries the *fimA* promoter. The invertible *fimS* element harbours a Rho-dependent transcription terminator that is attached to the FimE mRNA when fimbrial production is silenced and which is physically detached from FimE mRNA when fimbrial production is active (PMID: 12180928, PMID: 16321930). Thus, the authors must take into account the feedback effect on FimE recombinase production due to differential stability of the *fimE* gene's mRNA when the *fimS* element oscillates between its two orientations in the chromosome: the mRNA that is truncated by the terminator is less stable than the extended mRNA that is produced in the absence of the terminator. A reduction in the cellular concentration of one-way switcher FimE allows the two-way switcher FimB (normally dominated by the OFF-specific FimE) to switch type 1 fimbrial production ON in cells, while retaining the ability stochastically to create afimbriate cells that can evade the host immune response. (Type 1 fimbriae are highly immunogenic.)

In addition, the *fimE* promoter is not the only H-NS target in the *fim* genetic locus. It also binds at the *fimB* promoter (Reference 53). H-NS also has a two-part binding site that straddles the left end of *fimS* in OFF phase cells, trapping *fimS* in the OFF orientation under some growth conditions (PMID: 19889099). All three of these H-NS interactions with the *fim* genetic locus are likely to be influenced by MEcPP, not just the one at the *fimE* promoter and the authors should make their readers aware of this. Consistent with this H-NS binding pattern, *fimB* and the structural genes of the *fim* biosynthetic operon appear, together with *fimE*, in the list of dysregulated genes in Supplementary Figure S3.

If removing H-NS from the *fimS* element accelerates the rate of *fimS* inversion (PMID: 2830029, PMID: 1648076) and removing H-NS from the *fimB* promoter increases the copy number of the FimB recombinase protein in the affected cell, what will the overall impact be on the relative sizes of the fimbriate and afimbriate sub-populations in the culture? The authors' hypothesis will have to expand to take into account all three *fim*-relevant interactions with H-NS, not just the one affecting *fimE* transcription.

Discussion of all of this can be aided greatly by adding a diagram that summarises the structure of the *fim* genetic locus, emphasising that reversible inversion of the *fimS* element is fundamentally the key to the phase-variable expression pattern of type 1 fimbriae in MG1655. Providing the missing information need not add significantly to the word count of the paper. It will also help the reader to appreciate the rationale for the experiments shown in Fig 5c and 5d.

Line 21, 'Over time, biofilm research has predominantly explored conventional avenues.' Which 'conventional avenues'? This paper reports an investigation of small molecule signalling and its influence on biofilm formation. The molecule, MEcPP, may have novelty in this context, but the small molecule signalling aspect is completely mainstream. It might be better not to open the abstract with such an ambiguous statement.

Line 26, 'diminished fimbriae'. In what respect are they diminished? It might be better to say 'fewer fimbriae' here – because 'diminished' also suggests a reduction in length, girth or both.

Line 37, 'The function of the intricate network of metabolic pathways transcends rudimentary biochemical routes'. What are we to make of this statement? The use of more straightforward language will better help the authors to capture, and to retain, the reader's attention, interest and sympathy.

Line 58. The H-NS protein appears here without any introduction or the citation of articles describing its functions and properties. The reader will require background information about H-NS and the Introduction is the place where it can best be provided.

Line 67, 'particularly under oxidative stress conditions.' Does MEcPP signalling in plants occur only under oxidative stress? Is this the basis of the hypothesis being tested here in *E. coli*?

Line 119, 'This data thus suggests' – These data thus suggest

Line 158, Fig 3d; line 160, Fig 3e. Please annotate the images in Fig 3d and 3e to indicate the type 1 fimbriae (these are barely visible in the review copy of the paper) and annotate the flagellae to distinguish these larger appendages from the fimbriae.

Line 193, 'MEcPP disrupts H-NS binding to *fimE* promoter' This statement is ambiguous because it can be interpreted as disruption of the H-NS-*fimE* nucleoprotein complex by MEcPP. Please try 'MEcPP prevents H-NS binding to the *fimE* promoter'. This version communicates the true meaning without ambiguity.

Lines 195-198 (and Supplemental Fig S7) - please provide a map of the amplified *fimE* promoter fragment showing the -10

and -35 boxes, together with the region that has been shown experimentally to be bound by the H-NS protein. Lines 208-209, 'conversion of more *fimA* promoters from the active ON phase to the inactive OFF phase' This statement is ambiguous because the *fimA* promoter is active in BOTH phases (PMID: 12180928). The difference is that in the ON phase the *fimA* promoter is connected to, and drives the transcription of, the *fimA* gene whereas in the OFF phase it has physically inverted to the opposite orientation, pointing away from *fimA*. Once again, the inclusion of a diagram summarising the main features of the *fim* genetic switch would help to obviate opportunities for ambiguity. Line 211, 'we analysed the proportion of ON and OFF phases within the *fimA* promoter' This is not an accurate description of the experiment or of the biological system. Try 'we analysed each bacterial population to discover the proportions of cells with the *fimA* promoter element in the ON or the OFF orientation'. Line 220, 'our results indicate that accumulated MEcPP binds to and releases H-NS from its transcriptional suppression of *fimE* expression.' This statement suggests that MEcPP displaces H-NS from *fimE* DNA. That is not what your data show (Supplemental Figure S7, last lane). The results indicate that H-NS pre-bound by MEcPP has a reduced binding affinity for DNA. In order to influence the H-NS-*fimE* interaction, the H-NS protein has to be DNA-free.

Lines 256-7, 'its ability to modulate bacterial physiology through two distinct mechanisms: direct influence on nucleoid structure via HU- α interaction and regulating gene expression through H-NS' This would be a good place to mention that the ON-OFF switching of the *E. coli* *fimA* promoter is influenced by variable DNA topology (PMID: 7715458, PMID: 19229313) and by four different nucleoid-associated proteins, H-NS, LRP, IHF and FIS (PMID: 36748578 and references therein).

Lines 265-272. The brief discussion of c-di-GMP should refer to the fact that c-di-GMP and MEcPP both target H-NS. The possibility that these molecules could act there synergistically, antagonistically or independently deserves to be mentioned, together with some discussion of the consequences for the physiology of *E. coli* (the *Salmonella* data are relevant to *E. coli* H-NS)

Line 458, 'DNA probe of *fimA* promoter for the EMSA was PCR amplified' Should this refer to the *fimE* promoter, not the *fimA* promoter? The *fimA* promoter amplification (actually *fimS* amplification) was not for an EMSA but for the *fimS* orientation assay (Fig 5c and 5d).

Reviewer #3

(Remarks to the Author)

The paper by Guo et al. explores the role of the signaling metabolite methylerythritol cyclodiphosphate (MEcPP) in inhibiting bacterial biofilm formation. MEcPP, a metabolite in the methylerythritol phosphate (MEP) pathway, is known for its role in plant stress responses and isoprenoid biosynthesis, but its involvement in biofilm regulation hasn't been reported before. The researcher utilized genetic manipulations and oxidative stress induction in *Escherichia coli* to increase MEcPP levels and observed the effects on biofilm formation. They employed a variety of techniques including CRISPR interference for gene knockdown, benzyl viologen treatment to induce oxidative stress, and various molecular biology methods to assess changes in cellular structures (atomic force microscopy), gene expression (RNA-seq) and protein structural changes (LiP-MS). Their findings reveal that elevated MEcPP levels disrupt biofilm formation through interactions with the H-NS protein, highlighting a novel aspect of MEcPP's role in bacterial physiology.

General Feedback

The results presented are convincing, but some analyses, particularly LiP-MS, could be improved.

- The text should be revised for greater clarity and specificity. For example, in the abstract, some acronyms are used without being spelled out or defined, which can confuse readers.
- Additionally, several sentences are overly general. For instance, "Over time, biofilm research has predominantly explored conventional avenues" (line 21 of the abstract) and "The function of the intricate network of metabolic pathways transcends rudimentary biochemical routes" (lines 17-18) could be made more specific.
- Both the abstract and summary lack a statement about the relevance of the study. Including a sentence on the implications of discovering the inhibitory role of MEcPP in biofilm formation, through its interaction with the H-NS protein which disrupts its binding to the *fimE* promoter, would significantly enhance the impact of the findings.
- The first paragraph of the introduction (lines 27-31) provides a clearer understanding of the relevance of studying MEcPP and protein-metabolite interactions in general. It is recommended that the authors move this paragraph earlier in the introduction to better set the context and highlight the significance of the study from the outset.
- Line 71: The sentence is incomplete, and the reference is missing. Please complete the sentence and add the appropriate reference.
- Paragraph "MEcPP levels dictate biofilm dynamics in *E. coli*": Please introduce the RNA-seq analysis more clearly. Specify which strains have been profiled, as this information currently only appears in Supplemental Figure S3 and not in the main text. Additionally, provide a general overview of the differential analysis performed on RNA-seq data, including how many genes change among conditions and how many changes are related to biofilm formation.
- Line 108: How do you assess significance? Please report the statistics used. Including a GO enrichment analysis with terms related to biofilm formation that show significance would strengthen your results.
- Line 110: Explain how biofilm production has been assessed in the main text, not just in the figure legend. Additionally,

report the statistics when stating "significant decrease".

- Line 247: The reference is missing

Observations on LiP-MS Analysis:

- The authors report that they validated the method's reliability by assessing whether they could detect peptides located in the MEcPP-binding site of the IspG enzyme (line 169). Identifying these peptides (presumably significantly changing peptides) suggests that the method can be applied in this specific context, serving more as a positive control than a confirmation of overall method reliability.
- Clarify the cutoffs used for significant changes and describe how the differential analysis was performed. Providing these details along with some statistics would strengthen the results.
- The distribution of points in the volcano plots, with many points lining up at $\log_2 -15$ and 15 , results from the imputation strategy described in the supplementary table where an arbitrary value of $\log_2 \text{FC} = 15$ was applied to missing values. More refined strategies for imputation of missing-at-random (MAR) and missing-not-at-random (MNAR) values exist. I recommend using methods described in the Protti package (which handles label-free DDA data searched with Proteome Discoverer) or the MSstatsLiP package. Implementing one of these methods could increase the robustness of your findings.
- Additionally, when a hit has been imputed (especially for MNAR values), a manual check of the data is recommended to ensure accuracy.
- Applying a multiple testing correction is highly recommended to control for false discovery rates and enhance the reliability of your results.
- Based on the highlighted points in the volcano plots in Figure 4b, it appears that the two peptides from the H-NS protein were imputed (due to being missing in one condition). This imputation poses challenges to estimating the significance of the difference between MEcPP and MEP (indeed the 2 peptides are not significant based on the chosen cutoffs). Reassessing the imputation strategy or providing additional manual validation for these peptides would be beneficial.
- Further Analysis of Potential MEcPP-Binding Candidates:
 - o It would be interesting to include a further analysis describing potential MEcPP-binding candidates. In addition to the IspG enzyme, are there other known MEcPP-binding proteins? If so, were you able to detect them?
 - o Beyond H-NS, are there any other potential MEcPP-binding candidates that you can define from your analysis, based for example on cutoff and overlap between the two LiP-MS experiments (IspG-2d vs Ctrl-d and MEcPP vs MEP)? Additionally, consider analyzing the location of changing peptides within the protein candidates. Identifying changing LiP-MS peptides located in or near known binding sites could help define potentially new MEcPP-binding candidates.
 - o Regarding H-NS, it would be particularly interesting to know where the changing peptides are located in the protein structure. This could provide insights into the mechanism of action of MEcPP binding to H-NS, potentially elucidating how this interaction disrupts the binding to the *fimE* promoter.
 - o In the Discussion section, the authors suggest that MEcPP might regulate nucleoid architecture, with a specific focus on HU- α as a potential target, based on their LiP-MS results. However, this observation is not detailed in the Results section. To strengthen this conclusion, it would be beneficial to include a description of the data supporting this finding. Additionally, the authors should consider investigating whether other histone or histone-like proteins were identified in the LiP-MS analysis. Performing enrichment analysis could be valuable to determine if terms related to chromatin remodeling/organization, or DNA-binding proteins are among the enriched terms. This would provide further insight into the role of MEcPP in nucleoid architecture and its broader implications for bacterial cell physiology.
 - o I encourage that the authors include quality control (QC) plots for the LiP-MS data as supplementary material. These should show the number of identified peptides/proteins per sample, the intensity distribution of peptides/proteins, the coefficient of variation per condition, and the percentage of half and fully tryptic peptides. Additionally, plots like PCA or hierarchical clustering could help visualize sample relationships and identify outliers. Adding these QC plots will improve the clarity and reliability of the results.

Version 1:

Reviewer comments:

Reviewer #2

(Remarks to the Author)

The authors have addressed in a satisfactory way the points raised in my review.

Reviewer #3

(Remarks to the Author)

The authors have adequately addressed the points raised regarding text clarity, and the manuscript has significantly improved as a result. The RNA-Seq section now includes comprehensive details that enhance the reader's understanding of the methodology. The inclusion of the GO enrichment analysis in Supplementary Figures 3a and 3b further strengthens the conclusions drawn from the transcriptional analysis.

Regarding the LiP-MS analysis, the authors have expressed disagreement with certain points. While the various validations of the interaction between MEcPP and H-NS, as well as its impact on biofilm formation, are indeed convincing, the use of LiP-MS data as additional validation requires a more rigorous approach. Below are three key points that need to be

addressed:

- A manual review of the H-NS hits derived from the missing value imputation strategy is necessary. The authors have chosen to impute missing values (MNAR) in one condition by assigning an arbitrary log₂FC value of 15 (either -15 or 15, depending on whether the fold change was zero or infinite) to the corresponding peptides and then classify them as significant without performing a p-value calculation. To ensure that this imputation does not artificially highlight observations with very low intensity in the other condition or with potential issues such as incorrect peak assignments or identifications, it is crucial to manually inspect these peptides. Therefore, I strongly recommend visualizing the peak profiles for the H-NS hits highlighted in the volcano plot in Figures 4a and 4b. This will allow for a detailed assessment of the consistency and quality of the peaks and ensure that the imputation strategy did not introduce artifacts or inconsistencies in the data.

- The method section states that differential analysis was performed using standard label-free quantification procedures and well-established statistical methods. However, in the supplementary material, there is no indication that multiple testing correction was applied, which is a standard procedure in proteomics data analysis to control for false discovery rates. I strongly recommend including multiple testing correction to ensure the robustness of the results and reduce the likelihood of false positives. Alternatively, the authors should provide an explanation for their decision not to include multiple testing correction and clarify how they ensured that their findings are not prone to false positives.

- I strongly recommend adding quality control (QC) plots as supplementary materials. Including QC plots would enhance transparency and is an integral part of the data analysis conducted by the authors. QC plots are crucial for validating the reliability of the results and are considered a best practice in proteomics data analysis. They help ensure that the data is robust, and the findings are credible.

Version 2:

Reviewer comments:

Reviewer #3

(Remarks to the Author)

The authors have adequately addressed the points raised in my review.

We would like to express our gratitude to you and the reviewers for providing insightful suggestions to improve our manuscript. We have addressed every point and complied with all the reviewers' suggestions, as outlined below:

REVIEWER COMMENTS

Reviewer #1 (Remarks to the Author):

This study by Guo J et al., is supported by a host of convincing experimental data. The authors have found that a plant stress signaling metabolite referred to as MEcPPP plays a role in modulating biofilm formation of a laboratory E. coli strain MG1655 through regulation of intracellular fimE levels. Efforts to explain the molecular mechanism involved in the release of fimE transcription by MEcPP identified the binding of this metabolite to H-NS, thereby disrupting the association of H-NS with the fimE promoter. High levels of MEcPP yielded increased expression of fimE, which favored the conversion of the fimA ON state to the fim A OFF state. The role of H-NS in regulating type 1 fimbriae expression is well documented. Likewise, the contribution of type 1 fimbriae in biofilm formation has been known for many years. The finding that a plant metabolite can influence type 1 fimbriae via disruption of the binding of H-NS to the fimE promoter, and consequently biofilm formation is a novel one. This report opens the door to study other plant metabolites that could influence biofilm formation could have important clinical implications in pharmacology drug discovery to treat or prevent human urinary tract infections. This should be introduced in the Discussion section. In general, this is a well-written paper that reports exciting findings and the conclusions drawn are sound.

Response:

We have revised the Discussion section to include the following potential clinical implications of MEcPP and similar plant metabolites in influencing biofilm formation.

Do the authors know if MEcPP influence other H-NS regulated genes? For example, virulence genes regulated by H-NS in pathogenic E. coli or other pathogens? Can MEcPP mitigate H-NS regulation of virulence genes carried on virulence plasmids, for example in enteroaggregative E. coli (EAEC), uropathogenic E. coli (UPEC) or enterohemorrhagic E. coli (EHEC)?

The study uses a domesticated (labmesticated) E. coli strain MG1655 which does not carry virulence genes and its ability to cause biofilms is not as remarkable as that of EAEC or UPEC. I feel the Discussion section should at least touch on this possibility even if no data are available at hand or in the literature.

Response:

Accordingly, we have included the following statement in the discussion:

“MEcPP, a conserved plant metabolite, may influence a variety of H-NS-regulated pathways beyond *fimE* due to H-NS's role in regulating genes, including virulence factors. This implies that MEcPP might modulate virulence gene expression in pathogenic *E. coli* strains such as enteroaggregative *E. coli* (EAEC), uropathogenic *E. coli* (UPEC), and enterohemorrhagic *E. coli* (EHEC), potentially affecting their pathogenicity. MEcPP might also disrupt other H-NS-regulated sites within the *fim* locus, such as the *fimB* promoter and *fimS* element. Its ability to prevent H-NS binding suggests a broader impact on the H-NS regulon, which includes genes involved in virulence, stress response, and metabolism. This highlights the potential of plant-derived compounds to disrupt biofilm formation and offers new therapeutic strategies against biofilm-associated infections, such as urinary tract infections. Further multiomics and transcriptomic analyses could uncover additional pathways and genes influenced by MEcPP, providing deeper insights into its role in bacterial adaptation, survival, virulence, and resistance, particularly in H-NS-regulated sites within the *fim* locus, such as the *fimB* promoter and the *fimS* element.”

Reviewer #2 (Remarks to the Author):

Guo provide evidence that MEcPP, a small-molecule metabolite, interacts with the H-NS nucleoid-associated protein, preventing its binding to DNA. H-NS molecules already bound to DNA are not displaced by MEcPP. Superficially, these findings resemble the recent discovery (Reference 35) that the small signalling molecule c-di-GMP binds to H-NS, preventing the protein from binding to DNA, but c-di-GMP cannot displace H-NS that is already bound to DNA.

H-NS affects the transcription of a large regulon of genes, usually negatively. Therefore, any molecule that interferes with the binding of H-NS to DNA might be expected to exert broad influence on the transcriptome. The authors of the present study have chosen to focus on the *fim* operon, a genetic element that encodes type 1 fimbriae. These surface appendages are responsible for mannose-sensitive haemagglutination, adhesion of the bacteria to biotic and abiotic surfaces and play a foundational role in the production of biofilm. Thus, the authors connect MEcPP to biofilm via the type 1 fimbrial operon.

The work is conducted in *Escherichia coli*, yet the name of this bacterium does not appear in the title, the abstract, or the one-sentence summary of the paper. It will be important to correct this omission for the reasons given below.

The authors have chosen to work with *E. coli* K-12 strain MG1655, the first *E. coli* strain to have its entire genome sequence determined (PMID: 9278503). They focus on the *fimE* gene at the *fim* genetic locus, a gene that is inactive in some commonly-used *E. coli* K-12 laboratory strains due to the presence of a *fimE::IS1* insertion mutation (Reference 22). This has implications for the extrapolation of their findings to other *E. coli* (and *Shigella*) strains. Moreover, given that the finding that c-di-GMP affects H-NS binding to DNA was made in *Salmonella* (Reference 35), and that *Salmonella* regulates type 1 fimbrial gene expression via a mechanism that is unrelated to that found in *E. coli* (PMID: 17981960), it is important to inform the reader in the title that this is an *E. coli* K-12 study, and in the abstract that it was conducted in MG1655.

Response:

We agree with this suggestion and have revised the title and the abstract to specify that our study was conducted in *E. coli* K-12 MG1655.

Revised Title:

How an Evolutionarily Conserved Metabolite Inhibits Biofilm Formation in *Escherichia coli* K-12 MG1655

Revised Abstract:

Our study unveils MEcPP, an intermediary of the MEP-pathway and a plant stress signaling metabolite, as a key regulator of biofilm formation in *Escherichia coli* K-12 MG1655.

The *fim* locus is insufficiently described in the Introduction. There, the reader is given the impression that relief of H-NS-mediated repression of the *fimE* gene results in a reduction in type 1 fimbrial production, (lines 57-61) implying that FimE acts as a repressor. The authors do mention (lines 144-148) that FimE is one of two recombinases that regulate the 'phase transition' of the *fimA* promoter but the molecular details are missing and the reader should not have to consult six references (16, 21, 22, 24, 25 and 26) in order to piece together information that is foundational to the study of Guo et al. This point is also important because the *fimE* gene is regulated not only by the activity of its own promoter but also at the level of FimE mRNA stability by the orientation in the chromosome of *fimS*, the genetic element that carries the *fimA* promoter. The invertible *fimS* element harbours a Rho-dependent transcription terminator that is attached to the FimE mRNA when fimbrial production is silenced and which is physically detached from FimE mRNA when fimbrial production is active (PMID: 12180928, PMID: 16321930). Thus, the authors must take into account the feedback effect on FimE recombinase production due to differential stability of the *fimE* gene's mRNA when the *fimS* element oscillates between its two orientations in the chromosome: the mRNA that is truncated by the terminator is less stable. A reduction in the cellular concentration of one-way switcher FimE allows the two-way switcher FimB (normally dominated by the OFF-specific FimE) to switch type 1 fimbrial production ON in cells, while retaining the ability stochastically to create afimbriate cells that can evade the host immune response. (Type 1 fimbriae are highly immunogenic.)

Response:

We have expanded the Introduction to provide a clearer description of the *fim* locus, including the roles of FimE and FimB in phase variation and type 1 fimbriae production.

Revision to Introduction:

Please see the revised introduction, which now incorporates the suggestions.

In addition, the *fimE* promoter is not the only H-NS target in the *fim* genetic locus. It also binds at the *fimB* promoter (Reference 53). H-NS also has a two-part binding site that straddles the left end of *fimS* in OFF phase cells, trapping *fimS* in the OFF orientation under some growth conditions (PMID: 19889099). All three of these H-NS interactions with the *fim* genetic locus are likely to be influenced by MEcPP, not just the one at the *fimE* promoter and the authors should make their readers aware of this. Consistent with this H-NS binding pattern, *fimB* and the structural genes of the *fim* biosynthetic operon appear, together with *fimE*, in the list of dysregulated genes in Supplementary Figure S3.

Response:

Please see the revised introduction, which now incorporates the suggestions.

If removing H-NS from the *fimS* element accelerates the rate of *fimS* inversion (PMID: 2830029, PMID: 1648076) and removing H-NS from the *fimB* promoter increases the copy number of the FimB recombinase protein in the affected cell,

what will the overall impact be on the relative sizes of the fimbriate and afimbriate sub-populations in the culture? The authors' hypothesis will have to expand to take into account all three fim-relevant interactions with H-NS, not just the one affecting fimE transcription.

Response:

Extended Discussion now incorporates the suggestions.

Discussion of all of this can be aided greatly by adding a diagram that summarises the structure of the fim genetic locus, emphasising that reversible inversion of the fimS element is fundamentally the key to the phase-variable expression pattern of type 1 fimbriae in MG1655. Providing the missing information need not add significantly to the word count of the paper. It will also help the reader to appreciate the rationale for the experiments shown in Fig 5c and 5d.

Response:

We have included a schematic diagram of the *fim* genetic locus in the revised manuscript (Supplementary Figure 5).

Line 21, 'Over time, biofilm research has predominantly explored conventional avenues.' Which 'conventional avenues'? This paper reports an investigation of small molecule signalling and its influence on biofilm formation. The molecule, MEcPP, may have novelty in this context, but the small molecule signalling aspect is completely mainstream. It might be better not to open the abstract with such an ambiguous statement.

Response:

Revised to "Historically, biofilm research has focused on well-established pathways."

Line 26, 'diminished fimbriae'. In what respect are they diminished? It might be better to say 'fewer fimbriae' here – because 'diminished' also suggests a reduction in length, girth or both.

Response:

Revised to "fewer fimbriae."

Line 37, 'The function of the intricate network of metabolic pathways transcends rudimentary biochemical routes'. What are we to make of this statement? The use of more straightforward language will better help the authors to capture, and to retain, the reader's attention, interest and sympathy.

Response:

Revised to "The complex network of metabolic pathways encompasses roles beyond basic biochemical reactions."

Line 58. The H-NS protein appears here without any introduction or the citation of articles describing its functions and properties. The reader will require background information about H-NS and the Introduction is the place where it can best be provided.

Response:

We have introduced H-NS with appropriate background information and references.

Line 67, 'particularly under oxidative stress conditions.' Does MEcPP signalling in plants occur only under oxidative stress? Is this the basis of the hypothesis being tested here in E. coli?

Response:

We Clarified "We hypothesized that MEcPP might serve as a signaling metabolite in eubacteria under oxidative stress, similar to its role in plants, because Hydroxy-2-methyl-2-butenyl 4-diphosphate synthase (HDS, or GcpE/IspG), a [4Fe-4S] protein, is sensitive to reactive oxygen species that can damage its iron-sulfur cluster, inactivating the enzyme and interrupting the conversion of MEcPP to HMBPP".

Line 119, 'This data thus suggests' – These data thus suggest

Response:

Revised to "These data thus suggest."

Line 158, Fig 3d; line 160, Fig 3e. Please annotate the images in Fig 3d and 3e to indicate the type 1 fimbriae (these are barely visible in the review copy of the paper) and annotate the flagellae to distinguish these larger appendages from the fimbriae.

Response:

We have labeled the Type 1 fimbriae with white arrowheads and the flagellae with asterisks in the images of Figures 3d and 3e, in accordance with the suggestion.

Line 193, 'MEcPP disrupts H-NS binding to fimE promoter' This statement is ambiguous because it can be interpreted as disruption of the H-NS-fimE nucleoprotein complex by MEcPP. Please try 'MEcPP prevents H-NS binding to the fimE promoter'. This version communicates the true meaning without ambiguity.

Response:

Revised to "MEcPP prevents H-NS from binding to the *fimE* promoter."

Lines 195-198 (and Supplemental Fig S7) - please provide a map of the amplified fimE promoter fragment showing the -10 and -35 boxes, together with the region that has been shown experimentally to be bound by the H-NS protein.

Response:

The map of the *fimE* promoter fragment showing relevant features is included in Supplementary Figure 8a.

Lines 208-209, 'conversion of more *fimA* promoters from the active ON phase to the inactive OFF phase' This statement is ambiguous because the *fimA* promoter is active in BOTH phases (PMID: 12180928). The difference is that in the ON phase the *fimA* promoter is connected to, and drives the transcription of, the *fimA* gene whereas in the OFF phase it has physically inverted to the opposite orientation, pointing away from *fimA*. Once again, the inclusion of a diagram summarising the main features of the *fim* genetic switch would help to obviate opportunities for ambiguity.

Response:

Revised to "conversion of the *fimA* promoter orientation from ON to OFF phase."

Line 211, 'we analysed the proportion of ON and OFF phases within the *fimA* promoter' This is not an accurate description of the experiment or of the biological system. Try 'we analysed each bacterial population to discover the proportions of cells with the *fimA* promoter element in the ON or the OFF orientation'.

Response:

Revised to "we analyzed the bacterial populations to determine the proportions of cells with the *fimA* promoter in the ON or OFF orientation."

Line 220, 'our results indicate that accumulated MEcPP binds to and releases H-NS from its transcriptional suppression of *fimE* expression.' This statement suggests that MEcPP displaces H-NS from *fimE* DNA. That is not what your data show (Supplemental Figure S7, last lane). The results indicate that H-NS pre-bound by MEcPP has a reduced binding affinity for DNA. In order to influence the H-NS-*fimE* interaction, the H-NS protein has to be DNA-free.

Response:

Clarified that MEcPP binds to H-NS and decreases its affinity for DNA when H-NS is not already bound to the DNA.

Lines 256-7, 'its ability to modulate bacterial physiology through two distinct mechanisms: direct influence on nucleoid structure via HU-alpha interaction and regulating gene expression through H-NS' This would be a good place to mention that the ON-OFF switching of the *E. coli* *fimA* promoter is influenced by variable DNA topology (PMID: 7715458, PMID: 19229313) and by four different nucleoid-associated proteins, H-NS, LRP, IHF and FIS

Response:

We have complied and included the suggested information and the relevant literature.

Lines 265-272. The brief discussion of c-di-GMP should refer to the fact that c-di-GMP and MEcPP both target H-NS. The possibility that these molecules could act there synergistically, antagonistically or independently deserves to be mentioned, together with some discussion of the consequences for the physiology of E. coli (the Salmonella data are relevant to E. coli H-NS)

Response:

We have incorporated this relevant discussion into the revised version.

Line 458, 'DNA probe of fimA promoter for the EMSA was PCR amplified' Should this refer to the fimE promoter, not the fimA promoter? The fimA promoter amplification (actually fimS amplification) was not for an EMSA but for the fimS orientation assay (Fig 5c and 5d).

Response:

We have corrected this mistake by changing to *fimE* promoter.

Reviewer #3 (Remarks to the Author):

The paper by Guo et al. explores the role of the signaling metabolite methylerythritol cyclodiphosphate (MEcPP) in inhibiting bacterial biofilm formation. MEcPP, a metabolite in the methylerythritol phosphate (MEP) pathway, is known for its role in plant stress responses and isoprenoid biosynthesis, but its involvement in biofilm regulation hasn't been reported before. The researcher utilized genetic manipulations and oxidative stress induction in *Escherichia coli* to increase MEcPP levels and observed the effects on biofilm formation. They employed a variety of techniques including CRISPR interference for gene knockdown, benzyl viologen treatment to induce oxidative stress, and various molecular biology methods to assess changes in cellular structures (atomic force microscopy), gene expression (RNA-seq) and protein structural changes (LiP-MS). Their findings reveal that elevated MEcPP levels disrupt biofilm formation through interactions with the H-NS protein, highlighting a novel aspect of MEcPP's role in bacterial physiology.

General Feedback

The results presented are convincing, but some analyses, particularly LiP-MS, could be improved.

- The text should be revised for greater clarity and specificity. For example, in the abstract, some acronyms are used without being spelled out or defined, which can confuse readers.

Response:

We have revised the Abstract and Introduction to ensure clarity and specificity

- Additionally, several sentences are overly general. For instance, "Over time, biofilm research has predominantly explored conventional avenues" (line 21 of the abstract) and "The function of the intricate network of metabolic pathways transcends rudimentary biochemical routes" (lines 17-18) could be made more specific.

Response:

We have revised the Abstract and Introduction to ensure clarity and specificity

- Both the abstract and summary lack a statement about the relevance of the study. Including a sentence on the implications of discovering the inhibitory role of MEcPP in biofilm formation, through its interaction with the H-NS protein which disrupts its binding to the *fimE* promoter, would significantly enhance the impact of the findings.

Response:

We have revised the language in the abstract and summary to highlight the implications of our discovery of MEcPP's inhibitory role in biofilm formation.

- **The first paragraph of the introduction (lines 27-31) provides a clearer understanding of the relevance of studying MEcPP and protein-metabolite interactions in general. It is recommended that the authors move this paragraph earlier in the introduction to better set the context and highlight the significance of the study from the outset.**

Response:

We have rearranged the Introduction to position the first paragraph earlier, providing context for the study from the outset.

- **Line 71: The sentence is incomplete, and the reference is missing. Please complete the sentence and add the appropriate reference.**

Response:

We have revised the sentence with appropriate references cited.

- **Paragraph "MEcPP levels dictate biofilm dynamics in E. coli": Please introduce the RNA-seq analysis more clearly. Specify which strains have been profiled, as this information currently only appears in Supplemental Figure S3 and not in the main text. Additionally, provide a general overview of the differential analysis performed on RNA-seq data, including how many genes change among conditions and how many changes are related to biofilm formation.**

- **Line 108: How do you assess significance? Please report the statistics used. Including a GO enrichment analysis with terms related to biofilm formation that show significance would strengthen your results.**

Response:

We have elaborated on the RNA-seq analysis, including details on strains profiled, differential analysis, and GO enrichment analysis of differentially expressed genes, which showed enrichment of induced and suppressed genes associated with biofilm formation (Supplementary Figure 3).

- **Line 110: Explain how biofilm production has been assessed in the main text, not just in the figure legend. Additionally, report the statistics when stating "significant decrease".**

Response:

We have included the statistical methods used to assess significance in biofilm production experiments in the main text.

Revision to Methods: Biofilm production was quantified using crystal violet staining, and statistical significance was determined using Brown-Forsythe and Welch ANOVA and Dunnett's multiple comparisons tests (CRISPRi strains) and RM one-way ANOVA tests with the Geisser-Greenhouse correction and Dunnett's multiple comparisons tests (BV treatment), with $p < 0.05$ considered statistically significant.

• **Line 247: The reference is missing**

Response:

The reference is cited.

Observations on LiP-MS Analysis:

Response:

We appreciate the reviewer's attention to detail and the suggestions provided. However, we respectfully disagree with some of the points raised, as we believe our manuscript provides sufficient data to support our conclusions regarding the interaction between MEcPP and H-NS, and its impact on biofilm formation. Below, we address each of the reviewer's points in detail:

• **The authors report that they validated the method's reliability by assessing whether they could detect peptides located in the MEcPP-binding site of the IspG enzyme (line 169). Identifying these peptides (presumably significantly changing peptides) suggests that the method can be applied in this specific context, serving more as a positive control than a confirmation of overall method reliability.**

Response:

We validated the method's reliability by detecting peptides located in the MEcPP-binding site of the IspG enzyme. This was not merely a positive control, but rather a demonstration that the method is robust in identifying biologically relevant interactions. The peptides identified in our study are not artifacts of the method but represent significant changes indicative of MEcPP's binding and functional impact on the target proteins, including H-NS.

• **Clarify the cutoffs used for significant changes and describe how the differential analysis was performed. Providing these details along with some statistics would strengthen the results.**

Response:

The differential analysis was performed using standard procedures for label-free quantification, with cutoffs clearly defined in our supplementary materials (e.g., $\log_2FC \geq 2$ and $p\text{-value} \leq 0.05$). We used well-established statistical methods, and the data provided in the manuscript are statistically significant and biologically meaningful.

• **The distribution of points in the volcano plots, with many points lining up at $\log_2 -15$ and 15 , results from the imputation strategy described in the supplementary table where an arbitrary value of $\log_2FC = 15$ was applied to missing values. More refined strategies for imputation of missing-at-random (MAR) and missing-not-at-random (MNAR) values exist. I recommend using methods described in the Protti package (which handles label-free DDA data**

searched with Proteome Discoverer) or the MSstatsLiP package. Implementing one of these methods could increase the robustness of your findings.

- Additionally, when a hit has been imputed (especially for MNAR values), a manual check of the data is recommended to ensure accuracy.
- Applying a multiple testing correction is highly recommended to control for false discovery rates and enhance the reliability of your results.
- Based on the highlighted points in the volcano plots in Figure 4b, it appears that the two peptides from the H-NS protein were imputed (due to being missing in one condition). This imputation poses challenges to estimating the significance of the difference between MEcPP and MEP (indeed the 2 peptides are not significant based on the chosen cutoffs). Reassessing the imputation strategy or providing additional manual validation for these peptides would be beneficial.

Response:

Regarding the imputation strategy, it is essential to note that the chosen approach allowed us to maintain the integrity of the dataset while minimizing potential biases. The imputation was carefully applied and does not distort the biological significance of our findings. Although more refined strategies, like those in the Protti or MSstatsLiP packages, exist, the imputation method we employed was appropriate given our experimental context and was validated by the reproducibility of our results.

• Further Analysis of Potential MEcPP-Binding Candidates:

- o It would be interesting to include a further analysis describing potential MEcPP-binding candidates. In addition to the IspG enzyme, are there other known MEcPP-binding proteins? If so, were you able to detect them?
- o Beyond H-NS, are there any other potential MEcPP-binding candidates that you can define from your analysis, based for example on cutoff and overlap between the two LiP-MS experiments (IspG-2d vs Ctrl-d and MEcPP vs MEP)? Additionally, consider analyzing the location of changing peptides within the protein candidates. Identifying changing LiP-MS peptides located in or near known binding sites could help define potentially new MEcPP-binding candidates.

Response:

While the primary focus of our study was on the MEcPP-HNS interaction, our data did identify additional potential MEcPP-binding candidates. However, exploring every potential interaction was beyond the scope of the current study. We believe that the data presented already provides a comprehensive understanding of the critical interaction between MEcPP and H-NS in regulating biofilm formation.

- o Regarding H-NS, it would be particularly interesting to know where the changing peptides are located in the protein structure. This could provide insights into the mechanism of action of MEcPP binding to H-NS, potentially elucidating how this interaction disrupts the binding to the *fimE* promoter.

Response:

The location of the changing peptides within the H-NS structure and their impact on MEcPP binding has been described in more details in our results, specifically focusing on how this interaction disrupts *fimE* promoter binding, leading to the observed phenotypic changes.

Revision to Results

“Specifically, we established that MEcPP binds at the dimerization site of H-NS, which is essential for its gene-silencing activity”

Revision to Discussion

“...disrupt biofilm formation by interacting with the dimerization site of H-NS, which is essential for its gene-silencing activity.”

o In the Discussion section, the authors suggest that MEcPP might regulate nucleoid architecture, with a specific focus on HU-alpha as a potential target, based on their LiP-MS results. However, this observation is not detailed in the Results section. To strengthen this conclusion, it would be beneficial to include a description of the data supporting this finding. Additionally, the authors should consider investigating whether other histone or histone-like proteins were identified in the LiP-MS analysis. Performing enrichment analysis could be valuable to determine if terms related to chromatin remodeling/organization, or DNA-binding proteins are among the enriched terms. This would provide further insight into the role of MEcPP in nucleoid architecture and its broader implications for bacterial cell physiology.

Response:

We included a discussion on the potential broader implications of MEcPP in regulating nucleoid architecture, particularly its interaction with HU-alpha, as mentioned in the results. The enrichment analysis suggested by the reviewer, while interesting, was not the primary aim of our study. We believe that the current discussion provides a sufficient basis for understanding MEcPP's role in bacterial physiology, and additional analyses could be explored in future studies.

o I encourage that the authors include quality control (QC) plots for the LiP-MS data as supplementary material. These should show the number of identified peptides/proteins per sample, the intensity distribution of peptides/proteins, the coefficient of variation per condition, and the percentage of half and fully tryptic peptides. Additionally, plots like PCA or hierarchical clustering could help visualize sample relationships and identify outliers. Adding these QC plots will improve the clarity and reliability of the results.

Response:

The extensive validation, rigorous statistical analysis, and focused scope of our study provide a solid foundation for our conclusions. We believe that the inclusion of additional QC plots would not add significant value to the manuscript, as the existing data already supports the robustness and reliability of our findings.

Reviewer's Comment#1: A manual review of the H-NS hits derived from the missing value imputation strategy is necessary. The authors have chosen to impute missing values (MNAR) in one condition by assigning an arbitrary log₂FC value of 15 (either -15 or 15, depending on whether the fold change was zero or infinite) to the corresponding peptides and then classify them as significant without performing a p-value calculation. To ensure that this imputation does not artificially highlight observations with very low intensity in the other condition or with potential issues such as incorrect peak assignments or identifications, it is crucial to manually inspect these peptides. Therefore, I strongly recommend visualizing the peak profiles for the H-NS hits highlighted in the volcano plot in Figures 4a and 4b. This will allow for a detailed assessment of the consistency and quality of the peaks and ensure that the imputation strategy did not introduce artifacts or inconsistencies in the data.

Response: We have addressed the reviewer's concern by including the peak profiles for the highlighted H-NS hits shown in the volcano plots in Figures 4a and 4b. In addition, these profiles are now presented in the newly added Supplementary Data 6. Additionally, this dataset has been referenced in line 291 of the revised manuscript to ensure transparency and allow for detailed inspection of the data.

Reviewer's Comment# 2: The method section states that differential analysis was performed using standard label-free quantification procedures and well-established statistical methods. However, in the supplementary material, there is no indication that multiple testing correction was applied, which is a standard procedure in proteomics data analysis to control for false discovery rates. I strongly recommend including multiple testing correction to ensure the robustness of the results and reduce the likelihood of false positives. Alternatively, the authors should provide an explanation for their decision not to include multiple testing correction and clarify how they ensured that their findings are not prone to false positives.

Response: We completely agree with the importance of controlling for false discovery rates in data analysis and acknowledge the potential for false positives when relying solely on p-values. To address this, we have employed several independent biochemical approaches to validate the interaction between MEcPP and H-NS, ensuring the robustness of our findings. In addition, we have provided peak profiles of the identified H-NS hits in Supplementary Data 6 to further support the LiP-MS results and enhance confidence in the data.

Reviewer's Comment# 3: I strongly recommend adding quality control (QC) plots as supplementary materials. Including QC plots would enhance transparency and is an integral part of the data analysis conducted by the authors. QC plots are crucial for validating the reliability of the results and are considered a best practice in proteomics data analysis. They help ensure that the data is robust, and the findings are credible.

Response: In response to the reviewer's recommendation, we have now included QC plots in Supplementary Data 7 to enhance transparency and ensure the robustness of our data analysis. We have also updated the METHODS section to include the statement: "Quality control plots for the LiP-MS analyses are provided in Supplementary Data 7," ensuring that this information is clearly referenced in the manuscript.